# Efficient Differentiable Discovery of Causal Order

## Abstract

We introduce a differentiable and scalable regularizer for causal order, which can be integrated into gradient-based learning systems to inject causal inductive bias. Our approach builds on Intersort (Chevalley et al., 2025c), a recently proposed score-based method that infers causal orderings from interventional data. While effective, Intersort requires discrete combinatorial optimization, making it computationally expensive and non-differentiable. We address this limitation by relaxing the score with continuous optimization techniques, including differentiable sorting and ranking, yielding a differentiable surrogate objective. The resulting formulation can be used as a regularizer to encode prior knowledge about causal order—whether derived from interventions, domain heuristics, or learned proxies. As a proof of concept, we instantiate this regularizer using single-variable interventions and demonstrate significant improvements in causal discovery tasks across diverse synthetic datasets. Our work enables scalable, differentiable, and flexible integration of causal order into modern learning systems.

## 1 Introduction

Causal discovery is fundamental for understanding complex systems by identifying underlying causal relationships from data. It has significant applications across various fields, including biology (Meinshausen et al., 2016; Chevalley et al., 2025a;b), medicine (Feuerriegel et al., 2024), and social sciences (Imbens & Rubin, 2015), where causal insights inform decision-making and advance scientific knowledge. Traditionally, causal discovery has relied heavily on observational data due to the practical challenges and costs associated with conducting large-scale interventional experiments. However, observational data alone often necessitates strong assumptions about the data distribution to ensure identifiability beyond the Markov equivalence class (Spirtes et al., 2000; Shimizu et al., 2006; Hoyer et al., 2008).The growing availability of large-scale interventional datasets, particularly in genomics (Replogle et al., 2022; Datlinger et al., 2017; Dixit et al., 2016), provides a promising avenue for overcoming these challenges by revealing causal mechanisms through targeted manipulations.

Recently, Chevalley et al. (2025c) introduced Intersort, which uses the notion of *interventional faithfulness* and a score on causal orders, enabling the inference of causal orderings by comparing marginal distributions across observational and interventional settings. However, its reliance on a combinatorial optimization over the permutahedron renders it computationally expensive and non-differentiable. Scalability remains a challenge, making this method impractical for applications involving large numbers of variables—a common scenario in fields such as genomics (Replogle et al., 2022; Chevalley et al., 2025a) and neuroscience that involve up to tens of thousands of variables.

In this work, we introduce a differentiable and scalable formulation of the Intersort score by reparametrizing it in terms of a continuous potential function and leveraging differentiable sorting techniques such as the Sinkhorn operator (Cuturi, 2013). This formulation allows the causal order score to be optimized within gradient-based frameworks and, more importantly, used as a *differentiable regularizer* to enforce causal order constraints.

While our implementation uses interventional data to construct the regularizer, the formulation is more general: the score can encode *prior knowledge about causal order*, whether derived from interventions, heuristics, domain expertise, or proxy models. This flexibility makes our approach suitable for use in a wide range of downstream tasks, including neural structure learning, representation learning, and model regularization.

To demonstrate the utility of our approach, we incorporate the regularizer into a causal discovery algorithm. Experiments across diverse synthetic data regimes—including linear models, random Fourier features, gene regulatory networks (GRNs), and neural networks—show that the regularized model achieves superior performance compared to baselines such as GIES (Hauser & Bühlmann, 2012) and DCDI (Brouillard et al., 2020), particularly on complex, nonlinear settings like GRNs. Our formulation also scales efficiently to thousands of variables and retains robustness across noise types and data distributions.

This work contributes to the broader effort of embedding causal reasoning into machine learning. By enabling causal order to act as a flexible, differentiable *prior knowledge* regularization, our approach allows learning systems to incorporate partial or noisy information about causal structure—whether obtained through interventions, domain heuristics, domain expert knowledge or external models. In doing so, we move beyond purely associational representations toward models that align with causal mechanisms. This approach has the potential to improve model generalization, robustness to interventions, and interpretability across domains such as genomics, neuroscience, and reinforcement learning.

## 2 METHOD

### 2.1 DEFINITIONS AND ASSUMPTIONS

In this section, we introduce notations and definitions that are used throughout the paper inspired by Pearl (2009); Peters et al. (2017).

Let $(\mathcal{M}, d)$ be a metric space, and let $\mathcal{P}(\mathcal{M})$ denote the set of probability measures over $\mathcal{M}$. We define $D$ to be a statistical distance function $D : \mathcal{P}(\mathcal{M}) \times \mathcal{P}(\mathcal{M}) \to [0, \infty)$ that measures the divergence between probability distributions on $\mathcal{M}$.

Consider a set of $d$ random variables $\mathbf{X} = (X_1, X_2, \ldots, X_d)$ indexed by $V = \{1, 2, \ldots, d\}$, with joint distribution $P_{\mathbf{X}}$. We denote the marginal distribution of each variable as $P_{X_i}$ for $i \in V$. A causal graph is a tuple $\mathcal{G} = (V, E)$ of nodes and edges that form a Directed Acyclic Graph (DAG), where $V$ is the set of nodes (variables), and $E \subseteq V \times V$ is the set of directed edges representing causal relationships. An edge $(i, j) \in E$ indicates that variable $X_i$ is a direct cause of variable $X_j$. Let $\mathbf{A}^{\mathcal{G}}$ be the adjacency matrix of $\mathcal{G}$, where $\mathbf{A}^{\mathcal{G}}_{ij} = 1$ if $(i, j) \in E$, and $\mathbf{A}^{\mathcal{G}}_{ij} = 0$ otherwise. For each node $j \in V$, the set of parents $\mathrm{Pa}(j)$ consists of all nodes with edges pointing to $j$, i.e., $\mathrm{Pa}(j) = \{i \in V \mid (i, j) \in E\}$. We denote the set of descendants of node $i$ as $\mathrm{De}_{\mathcal{G}}(i)$, which includes all nodes reachable from $i$ via directed paths. Similarly, the set of ancestors of $i$ is denoted as $\mathrm{An}_{\mathcal{G}}(i)$.

An SCM $\mathcal{C} = (\mathbf{S}, P_N)$ consists of a set of structural assignments $\mathbf{S}$ and a joint distribution over exogenous noise variables $P_N$. Each variable $X_j$ is assigned via a structural equation:

$$X_j = f_j \left( \mathbf{X}_{\mathrm{Pa}(j)}, N_j \right),$$

where $N_j$ is an exogenous noise variable, and $\mathbf{X}_{\mathrm{Pa}(j)}$ are the parent variables of $X_j$. The exogenous variable need not be independent, potentially introducing confounding. Revision: We denote the distribution induced by an SCM as $P_{X_j}^{\mathcal{C}, (\emptyset)}$ (observational distribution).

In our work, we focus on interventions that modify the structural assignments of certain variables. Specifically, we consider interventions where the structural assignment of a variable $X_k$ is replaced by a new exogenous variable $\tilde{N}_k$, independent of its parents $X_k = \tilde{N}_k$. Revision: We denote this distribution as $P_{X_j}^{\mathcal{C}, do(X_i := \tilde{N}_i)}$.

**Definition 2.1.** A *causal order* of the graph $\mathcal{G} = (V, E)$ is a permutation $\pi : V \to V$ such that for any edge $(i, j) \in E$, we have $\pi(i) < \pi(j)$. This ensures that causes precede their effects in the ordering (Peters et al., 2017). Multiple causal orders may satisfy the same DAG.

Since $\mathcal{G}$ is acyclic, at least one causal order exists, though it may not be unique. We denote the set of all valid causal orders consistent with $\mathcal{G}$ as $\Pi^*$.

**Definition 2.2.** To measure the discrepancy between a proposed permutation $\pi$ and the causal structure of the graph $\mathcal{G}$, we use the *top order divergence* (Rolland et al., 2022), defined as:

$$D_{\mathrm{top}}(\mathcal{G}, \pi) = \sum_{\pi(i) > \pi(j)} \mathbf{A}^{\mathcal{G}}_{ij}.$$

This divergence counts the number of edges that are inconsistent with the ordering $\pi$, i.e., edges where the cause appears after the effect in the proposed ordering. For any causal order $\pi^* \in \Pi^*$, we have $D_{\text{top}}(\mathcal{G}, \pi^*) = 0$.

**Assumption 2.3** (Interventional Faithfulness)**.** Interventional faithfulness (Chevalley et al., 2025c) assumes that all directed paths in the causal graph manifest as significant changes in the distribution under Revision: single variable interventions as measured by a statistical distance. Specifically, if intervening on variable $X_i$ leads to a detectable change in the distribution of variable $X_j$, then there must be a directed path from $X_i$ to $X_j$ in the causal graph $\mathcal{G}$. Conversely, if there is no directed path from $X_i$ to $X_j$, then intervening on $X_i$ does not affect the distribution of $X_j$ beyond a significance threshold $\epsilon$. Revision: See Definition 2.4 for a formal definition below.

Interventional faithfulness allows us to use statistical divergences between marginal observational and interventional distributions to infer the causal ordering of variables. By assuming interventional faithfulness, we can relate changes observed under interventions to the underlying causal structure. Importantly, this assumption does not require causal sufficiency: it is preserved under marginalization, so if the full system (including latent variables) is $\epsilon$-interventionally faithful, then the induced marginal distributions over the observed variables remain $\epsilon$-interventionally faithful (Chevalley et al., 2025c). This implies that our method can recover the correct causal order on the DAG component of an ADMG, even in the presence of unobserved confounding. More formally, it is defined as:

**Definition 2.4** (Chevalley et al. (2025c))**.** Given the distributions $P_X^{\mathcal{C},(\emptyset)}$ and $P_X^{\mathcal{C},do(X_k:=\tilde{N}_k)}, \forall k \in \mathcal{I}$, we say that the tuple $(\tilde{N}, \mathcal{C})$ is $\epsilon$-*interventionally faithful* to the graph $\mathcal{G}$ associated to $\mathcal{C}$ if for all $i \neq j, i \in \mathcal{I}, j \in V, D\left(P_{X_j}^{\mathcal{C},(\emptyset)}, P_{X_j}^{\mathcal{C},do(X_i:=\tilde{N}_i)}\right) > \epsilon$ if and only if there is a directed path from $i$ to $j$ in $\mathcal{G}$.

## 2.2 Differentiable score

Intersort has recently shown strong performance in recovering causal orderings from interventional data, but its scalability is limited. The algorithm relies on discrete local search over permutations, and our experiments confirm that runtimes grow steeply with the number of variables: while accuracy remains competitive up to a few dozen nodes, the method quickly becomes impractical beyond $d \approx 50$–$60$ (see Appendix J.1). This restricts its applicability to small or medium-scale settings, whereas many domains of interest—such as genomics (Replogle et al., 2022) or climate science (Nowack et al., 2020; Runge et al., 2019; Kalnay et al., 1996)—require reasoning over hundreds or thousands of variables. To overcome these limitations, we introduce a differentiable relaxation of the Intersort score. This formulation, which we call DIFFINTERSORT, scales efficiently to substantially larger graphs and, crucially, can be used as a regularizer within end-to-end gradient-based training pipelines. We first recall the discrete score underlying Intersort, and then present its differentiable extension.

*Intersort score–* Given an observational distribution $P_X^{\mathcal{C},(\emptyset)}$ and a set of interventional distributions $\mathcal{P}_{int} = \{P_X^{\mathcal{C},do(X_k:=\tilde{N}_k)}, k \in \mathcal{I}\}, \mathcal{I} \subseteq V$, Chevalley et al. (2025c) define the following score for a permutation $\pi$, for some statistical distance $D : \mathcal{P}(M) \times \mathcal{P}(M) \to [0, \infty), \epsilon > 0, c > \epsilon$:

$$S(\pi, \epsilon, D, \mathcal{I}, P_X^{\mathcal{C},(\emptyset)}, \mathcal{P}_{int}, c) = \sum_{\pi(i)<\pi(j),i\in\mathcal{I},j\in V} \left( D\left(P_{X_j}^{\mathcal{C},(\emptyset)}, P_{X_j}^{\mathcal{C},do(X_i:=\tilde{N}_i)}\right) - \epsilon \right)$$
$$+ c \cdot d \cdot \mathbf{1}_{D\left(P_{X_j}^{\mathcal{C},(\emptyset)}, P_{X_j}^{\mathcal{C},do(X_i:=\tilde{N}_i)}\right)>\epsilon} \tag{1}$$

Intuitively, the summation measures how well the causal order aligns with strong causal effects. The second term's rescaling by a factor of $d$ ensures that effects exceeding $\epsilon$ will prioritize ordering constraints, enforcing $\pi(i) < \pi(j)$ by amplifying their relative importance compared to effects smaller than $\epsilon$.

*Theoretical guarantees–* One of the key aspects of the Intersort score is that Chevalley et al. (2025c) were able to prove strong guarantees on the expected error of the optimal solution $\pi_{opt} = \arg\max_\pi S(\pi, \epsilon, D, \mathcal{I}, P_X^{\mathcal{C},(\emptyset)}, \mathcal{P}_{int}, c)$. Revision: As such, it informs on the ideal performance that can be reached if an algorithm finds a solution close to the optimum. More specifically, they derive an upper-bound on the error $\mathbb{E}[D_{top}(\mathcal{G}, \pi_{opt})]$ under a uniform distribution of the intervened variables, that is, the probability that $i \in V$ is in $\mathcal{I}$ is uniform, $p_{int} := P(i \in \mathcal{I}), \forall i \in V$. For example, for an Erdős–Rényi graph distribution, they show that the expected number of

misoriented edges scales with $\mathcal{O}(d)$. We thus aim to develop a scalable and differentiable score that retains the desirable guarantees on the optimum of Equation (1). Revision: We recall the main two theorems and their statements in Appendix D.

***Optimization Strategy–*** Intersort optimization consists of two steps: (1) an initial ordering via SOR-TRANKING, and (2) refinement via LOCALSEARCH. The first step sorts distances and constructs a topological order greedily, while the second iteratively refines the order in a local neighborhood. While effective, these steps have complexity of approximately $\mathcal{O}(d^3)$. We propose a differentiable alternative using continuous relaxation, replacing discrete permutations with a potential-based parameterization. Details on the original Intersort optimization are provided in Appendix E.

***DiffIntersort score–*** To make Intersort differentiable, we reparameterize the ordering of variables using a potential vector $\boldsymbol{p} \in \mathbb{R}^d$, where the relative values of $\boldsymbol{p}$ determine the causal ordering. Formally, the potential can be viewed as a scalar function $p : V \to \mathbb{R}$ assigning a value to each node $i \in V$, but in practice we represent it as a vector $\boldsymbol{p} = (p_1, \ldots, p_d)$ for convenience. Intuitively, $\boldsymbol{p}$ assigns a "height" to each variable, so that edges are directed from higher to lower potential values. This is analogous to a potential field in physics, where flows move downhill, and ensures that the induced graph is acyclic. From $\boldsymbol{p}$, we construct a soft permutation matrix using the Sinkhorn operator, which allows us to optimize continuously while maintaining differentiability. Formally, the permutation $\pi$ of the variables is represented by $\boldsymbol{p}$ such that $\pi(i) < \pi(j) \iff p_i > p_j$. The associated permutation matrix $\boldsymbol{\sigma}(\boldsymbol{p})$ is a $d \times d$ binary matrix with $\boldsymbol{\sigma}(\boldsymbol{p})_{ij} = 1$ if $\pi(i) = j$. We define a discrete gradient operator $(\mathrm{grad}(\boldsymbol{p}))_{ij} = p_i - p_j$, which encodes the pairwise precedence implied by the potential: it is nonnegative if and only if $i$ precedes $j$ in the order. Applying the element-wise $\mathrm{Step}$ function produces $(\mathrm{Step}(\mathrm{grad}(\boldsymbol{p})))_{ij} = \mathbf{1}_{p_i - p_j > 0}$, a binary matrix of possible edges consistent with $\boldsymbol{p}$.

We aim to rewrite the score such that it is parameterized by the potential $\boldsymbol{p}$. Building the matrix $\mathbf{D} \in \mathbb{R}^{d \times d}$ as

$$\mathbf{D}_{ij} = \begin{cases} D\left(\left(P_{X_j}^{\mathcal{C},(\emptyset)}, P_{X_j}^{\mathcal{C}, do(X_i := \tilde{N}_i)}\right) - \epsilon\right) + c \cdot d \cdot \mathbf{1}_{D\left(P_{X_j}^{\mathcal{C},(\emptyset)}, P_{X_j}^{\mathcal{C}, do(X_i := \tilde{N}_i)}\right) > \epsilon} & \text{if} \quad i \in \mathcal{I} \\ 0 & \text{if} \quad i \notin \mathcal{I} \end{cases} \tag{2}$$

we can write the score in terms of the potential instead of permutation as follows:

$$S(\boldsymbol{p}, \epsilon, D, \mathcal{I}, P_X^{\mathcal{C},(\emptyset)}, \mathcal{P}_{int}, c) = \langle \mathbf{D}, \mathrm{Step}(\mathrm{grad}(\boldsymbol{p})) \rangle_F. \tag{3}$$

The relationship between the potential and permutation is clarified through the following theoretical result.

**Theorem 2.5.** *Let* $\mathbb{P} = \arg\max_{\boldsymbol{p}} S(\boldsymbol{p}, \epsilon, D, \mathcal{I}, P_X^{\mathcal{C},(\emptyset)}, \mathcal{P}_{int}, c)$ *s.t.* $\boldsymbol{p}_i \neq \boldsymbol{p}_j \forall i, j \in V$, *be the set of potentials that maximize the score, such that no two entries of the potentials are equal.* $\Pi = \arg\max_{\pi} S(\pi, \epsilon, D, \mathcal{I}, P_X^{\mathcal{C},(\emptyset)}, \mathcal{P}_{int}, c)$ *be the set of permutations that maximize the Intersort score. For every* $\pi \in \Pi$, *there is a set* $\bar{p} \subset \mathbb{P}$ *such that* $\forall \boldsymbol{p} \in \bar{p} : \pi(i) < \pi(j) \iff p_i > p_j$.

The proof can be found in the appendix in Appendix G. This score is still not practically useful as it provides non-informative gradients for $\boldsymbol{p}$.

To obtain a differentiable relaxation, we approximate the permutation induced by the potential $\boldsymbol{p}$ using the Sinkhorn operator, which maps a score matrix $\boldsymbol{p}\boldsymbol{o}^\top$ into the Birkhoff polytope of doubly stochastic matrices. This provides a smooth surrogate for a permutation matrix that becomes exact as the temperature $t \to 0$ and the number of iterations $T \to \infty$. In practice, we use Revision: positive $t$ and finite $T$, yielding a differentiable approximation that can be optimized with gradient descent while remaining consistent with the discrete causal ordering. Substituting this relaxation into Equation (2) yields the DiffIntersort score:

$$S(\boldsymbol{p}) = \left\langle \mathbf{D}, \left(\mathcal{S}_{\mathrm{bin}}^T\left(\frac{\boldsymbol{p}\boldsymbol{o}^\top}{t}\right) \boldsymbol{L} \mathcal{S}_{\mathrm{bin}}^T\left(\frac{\boldsymbol{p}\boldsymbol{o}^\top}{t}\right)^\top\right) \right\rangle_F, \tag{4}$$

where $\mathcal{S}_{\mathrm{bin}}^T$ denotes the relaxed permutation projected back to a binary matrix via projection with the Hungarian algorithm Kuhn (1955), and $\boldsymbol{L}$ is an upper-triangular matrix of 1's. $\mathcal{S}_{\mathrm{bin}}^T$ is thus the permutation matrix associated to $\boldsymbol{p}$, and when applying this permutation to $\boldsymbol{L}$, we obtain the same matrix as $\mathrm{Step}(\mathrm{grad}(\boldsymbol{p}))$ (for $t \to 0$ and $T \to \infty$). Intuitively, this replaces the discrete ordering with a differentiable approximation, enabling gradient-based optimization. Further details

on the equivalence between permutation formulations, entropy regularization, and the projection are provided in Appendix A.

However, it is well established that gradient descent is not guaranteed to converge to an optimal solution. Particularly in the differentiable causal discovery setting, lack of convexity of the loss function constitutes an important factor in the failure of differentiable models, which depending on the data distributions can get stuck in local minima (Ng et al., 2024). Converge guarantees as well as lack of local optima are thus important properties for a differentiable causal structure learning algorithm, even though they are usually difficult or impossible to establish. Below, we thus analyze the optimization properties of our proposed score.

***Lipschitz Continuity of the DiffIntersort gradient–*** A key challenge in differentiable causal discovery is ensuring stable optimization. We establish that the gradient of the DiffIntersort score is Lipschitz continuous:

**Theorem 2.6** (Lipschitz continuity of $\nabla f$ with monotone dependence on $d$ and $t$). *Let $t > 0$ and let $\mathcal{B} \subset \mathbb{R}^{d \times d}$ denote the Birkhoff polytope. For $M \in \mathbb{R}^{d \times d}$ define the entropically regularized doubly-stochastic projection*

$$S(M) \in \arg\max_{P \in \mathcal{B}} \left\{ \langle P, M \rangle_F + t\, H(P) \right\}, \qquad H(P) = -\sum_{i,j} P_{ij} \log P_{ij}.$$

*Fix $D, L \in \mathbb{R}^{d \times d}$ and $o \in \mathbb{R}^d$, and for $p \in [-1, 1]^d$ set $M(p) = p\, o^\top$ and*

$$f(p) = \left\langle D,\ S(M(p))\, L\, S(M(p))^\top \right\rangle_F.$$

*Assume the uniform bounds*

$$\|o\|_\infty \le B_o, \qquad \|D\|_2 \le B_D, \qquad \|L\|_2 \le B_L.$$

*Then $f \in C^1([-1, 1]^d)$ and $\nabla f$ is Lipschitz on $[-1, 1]^d$, i.e.,*

$$\|\nabla f(p_1) - \nabla f(p_2)\|_2 \le L_f \|p_1 - p_2\|_2 \qquad \text{for all } p_1, p_2 \in [-1, 1]^d,$$

*with a constant of the form*

$$L_f \le C_0\, B_D\, B_L\, B_o^2\, \Phi\!\left(\frac{B_o}{t}\right) \Psi(d), \tag{5}$$

*where $C_0 > 0$ is a universal numerical constant, $\Phi : [0, \infty) \to [1, \infty)$ is nondecreasing, and $\Psi : \mathbb{N} \to [1, \infty)$ is nondecreasing. In particular, $L_f$ increases when $t$ decreases (through $\Phi(B_o/t)$) and when $d$ increases (through $\Psi(d)$).*

The proof can be found in Appendix G. This property ensures that gradient updates remain stable across iterations, preventing extreme fluctuations that could hinder convergence. In practice, this allows DiffIntersort to achieve smooth and consistent optimization, reducing the risk of getting stuck in poor local minima, particularly in high-dimensional settings. See Appendix F for a discussion of the optimization properties of the DiffIntersort score.

***Initial solution close to the optimum–*** Efficient optimization, particularly using gradient-based methods, critically depends on the quality of the initial solution. Starting from a point in the vicinity of the optimal set can significantly enhance convergence rates and ensure that the algorithm avoids undesirable local minima or saddle points. To this end, we define a task-dependent subset of the potential space informed by the distance matrix $\boldsymbol{D}$, which facilitates both theoretical analysis and practical performance.

**Definition 2.7.** We define $\boldsymbol{P}^{\mathbf{D}} := \left\{ \boldsymbol{p} \in [-1, 1]^d : \forall (i, j) \text{ s.t. } \mathbf{D}_{ij} > 0, p_i > p_j \right\}$, $\boldsymbol{P} \subseteq [-1, 1]^d$ the set of potentials whose order respect the causal effects greater than $\epsilon$ encoded in $\mathbf{D}$.

**Proposition 2.8.** *The set $\boldsymbol{P}^{\mathbf{D}} \subseteq [-1, 1]^d$ is convex.*

**Theorem 2.9.** *Assume that the data is $\epsilon$-interventional faithful. Let $\boldsymbol{p}^* \in \arg\max_{[-1,1]^d} \lim_{t \to 0} f(\boldsymbol{p}) = \boldsymbol{P}^*$. Then $\boldsymbol{P}^* \subseteq \boldsymbol{P}^{\mathbf{D}}$ and $\forall \boldsymbol{p} \in \boldsymbol{P}^{\mathbf{D}}$, $|\lim_{t \to 0} f(\boldsymbol{p}) - \lim_{t \to 0} f(\boldsymbol{p}^*)| \le \epsilon \frac{d(d-1)}{2}$, where $\lim_{t \to 0}$ denotes the score of a potential for the associated exact, binary permutation matrix.*

**Lemma 2.10.** *Assume that the data is $\epsilon$-interventional faithful. Then the SORTRANKING algorithm of Chevalley et al. (2025c) finds a potential $p \in P^{\mathbf{D}}$.*

The proofs can be found in Appendix G. The above results establish that the convex set $P^{\mathbf{D}}$ not only contains all limiting optimal solutions $P^*$ but also ensures that any potential within $P^{\mathbf{D}}$ is near-optimal up to a bounded error term dependent on $\epsilon$ and the dimensionality $d$, but independent of the task dependent matrix $\mathbf{D}$. The convexity of $P^{\mathbf{D}}$ (Proposition 2.8) facilitates the use of projection-based optimization techniques, guaranteeing that iterates remain within a region that respects the causal ordering encoded by $\mathbf{D}$. However, in practice, we find that no projection steps are needed, as appropriately scaling the factor $c$ in Equation (2) prevents a gradient iteration from pushing the solution out of $P^{\mathbf{D}}$.

Moreover, the SORTRANKING algorithm (Lemma 2.10) provides a practical method for generating an initial potential within $P^{\mathbf{D}}$, thereby ensuring that the optimization process begins in a favorable region. This proximity to the optimum allows gradient-based methods to leverage local smoothness properties, such as those guaranteed by the Łojasiewicz inequality, to achieve faster convergence rates and robust performance.

***Computational Complexity of DiffIntersort–*** DiffIntersort scales efficiently compared to combinatorial search methods. Its primary computational cost comes from (1) *Differentiable Score Computation:* $O(d^2)$ due to matrix operations; (2) *Sinkhorn Operator:* $O(d^2T)$, where $T$ is the number of iterations (typically $T = 500$); (3) *Hungarian Algorithm:* Worst-case $O(d^3)$, but empirically $O(d^2)$ due to initialization with a near-optimal solution. Thus, the practical complexity is closer to $O(d^2T)$ per gradient descent iteration, significantly outperforming combinatorial methods like Intersort. Experiments confirm scalability to thousands of variables (Figure 1 and Appendix J.1), making DiffIntersort well-suited for genomics and other high-dimensional applications.

### 2.3 DIFFINTERSORT AS A CAUSAL REGULARIZER

The DiffIntersort score provides a differentiable measure of how well a candidate ordering aligns with interventional data. While originally introduced for causal order recovery (Chevalley et al., 2025c), here we propose a new use: employing DiffIntersort as a *causal regularizer*. To our knowledge, this is the first work to explicitly formulate causal order regularization as a differentiable penalty that can be integrated into learning systems. To assess its potential, we use a simple differentiable model as a testbed. The central question is whether injecting ordering information through $S(\boldsymbol{p})$ improves structure recovery when combined with a standard fitting loss. We do not propose a new causal discovery algorithm *per se*, but rather use a simple differentiable model as a testbed to demonstrate the usefulness of causal regularization with DiffIntersort as a general principle.

Concretely, let $\mathbf{X} \in \mathbb{R}^{n \times d}$ be a dataset of $n$ observations of $d$ variables $\{X_1, \ldots, X_d\}$ under both observational and interventional conditions. A regularized causal discovery objective can then be formulated as

$$\min_{\theta, \mathbf{p}} \quad \mathcal{L}_{\text{fit}}(\theta, \mathbf{p}) Revision: -\lambda S(\mathbf{p}), \tag{6}$$

where $\theta$ are the parameters of the causal mechanisms (e.g., weights in a linear model), $\mathcal{L}_{\text{fit}}(\theta, \mathbf{p})$ is the fitting loss, and $\lambda > 0$ balances data fit and structural regularization. The regularizer thus encourages $\mathbf{p}$ to induce an ordering consistent with the interventional evidence.

For illustration, we adopt a linear SEM parameterization, where

$$X_j = \sum_{i=1}^{d} W_{ji} X_i + b_j + N_j, \tag{7}$$

with weight matrix $\mathbf{W} \in \mathbb{R}^{d \times d}$, bias $b_j$, and noise $N_j$. The causal ordering $\mathbf{p}$ induces a binary mask $M_{\mathbf{p}}$ that enforces acyclicity by restricting edges to flow only from earlier to later variables. The effective weights are then given by $\tilde{\mathbf{W}} = \mathbf{W} \circ M_{\mathbf{p}}^T$. When evaluating interventional environments, we additionally mask the intervened variable from its own prediction loss, ensuring consistency with structural constraints. Revision: We adopt the linear parameterization even when the data was generated from non-linear mechanisms.

This linear formulation is chosen solely to isolate the impact of DiffIntersort regularization. More complex parameterizations (e.g., neural models or permutahedron-based methods) could be substituted,

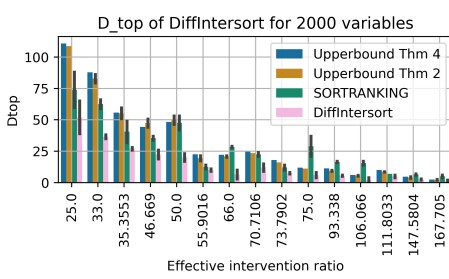 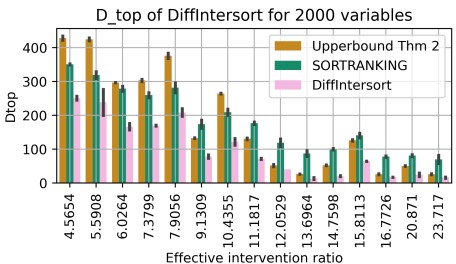

(a) Simulation ER with 2000 variables      (b) Simulation SF with 2000 variables

Figure 1: Simulation and comparison between the bounds of Theorems 2 and 4 of Chevalley et al. (2025c) for Erdős-Rényi (ER, left) and scale-free networks (SF, right) with 2000 variables. We compare the causal order obtained by maximizing our proposed DiffIntersort score and the output of SORTRANKING. For each setting, we draw Revision: 2 graphs per setting, following an ER distribution with a probability of edges per variable $p_e$ in $\{0.0001, 0.00005, 0.00002\}$ and following a Barabasi-Albert SF distribution, with an average edge per variable in $\{1, 2, 3\}$. A setting is the tuple $(p_{int}, p_e)$, where $p_e = \frac{2E(\#\text{edges})}{d(d-1)}$ for the SF distribution. For each graph, we run the algorithm on 1 configuration, where each configuration corresponds to a draw of the targeted variables following $p_{int}$. We have $p_{int} \in \{0.25, 0.33, 0.5, 0.66, 0.75\}$. Settings are ordered on the x-axis following the effective intervention ratio $\frac{p_{int}}{\sqrt{p_e}}$ (Chevalley et al., 2025c). Revision: The error bars show the 95% confidence interval around the mean.

but would confound the analysis by introducing additional sources of performance variation. Revision: Examples of existing methods the regularizer may be applied to can be found in Appendix C.3. For completeness, the full training algorithm and implementation details are provided in Appendix B.

## 3 EMPIRICAL RESULTS

We next evaluate the proposed DiffIntersort differentiable score both in its effectiveness in deriving the causal order of a system, as well as it usefulness as a differentiable regularizer in a causal discovery model.

We first evaluate the DiffIntersort score in it ability to recover the causal order in simulated graphs and distance matrices. We here reproduce the experiment of (Chevalley et al., 2025c). We compare the top order divergence of DiffIntersort to SORTRANKING, and to Intersort for 5 and 30 variables, and the upper-bounds of Thm 2 and Thm 4 derived in (Chevalley et al., 2025c) Revision: (see Appendix D). The upper bounds act as a sanity check, providing a measure of how close the approximate solution is to the true optimum of the score. We evaluate on both Erdős-Rényi distribution (Erdős et al., 1960) and scale-free network modeled by the Barabasi-Albert distribution Albert & Barabási (2002), with varying edge densities and intervention coverage. The results are reported in Figure 1 for 2000 variables and in Figures 6 and 7 for 5, 30, 100 and 1000 variables. It is crucial that our score be optimizable up to at least 2000 variables, as it is a common scale in real world datasets such as single-cell transcriptomics (Replogle et al., 2022). As studied previously, we initialize DiffIntersort with the solution of SORTRANKING, and use Adam (Kingma, 2014) to optimize the score. As observed, DiffIntersort fulfills the upper-bounds for all settings, even at large scale, which allows us to not reject the hypothesis that DiffIntersort finds an optimum of the score. At $d = 2000$, DiffIntersort matches the theoretical bounds closely and outperforms SORTRANKING across ER and SF graphs (Figure 1). Appendix Figures 6 and 7 show the same pattern at smaller scales, with variance bands across settings. Those results validate our proposed approach of solving the Intersort problem in a continuous and differentiable framework, and guarantees that it is not limited by scale. We also evaluate the scalability of DIFFINTERSORT against INTERSORT across increasing problem dimensions (see Appendix J.1). As shown in Figure 3, the runtime of INTERSORT grows rapidly with $d$, while DIFFINTERSORT scales more gently and benefits further from GPU acceleration. Accuracy results in Figure 4 show a similar pattern: both methods perform comparably at small scale, but DIFFINTERSORT achieves consistently lower top-order divergence for $d \gtrsim 50$, highlighting not only runtime gains but also accuracy advantages from differentiable optimization at larger $d$.

We now evaluate our method, DiffIntersort, on simulated data and compare its performance to various baseline methods. We follow the experimental setup of Chevalley et al. (2025c) to ensure a fair and consistent evaluation across different domains. See Appendix H for details about the synthetic data generation. Specifically, we generate graphs from an Erdős-Rényi distribution (Erdős et al., 1960) with an expected number of edges per variable $c \in \{1, 2\}$. Data is simulated using both linear relationships and random Fourier features (RFF) additive functions to capture non-linear dependencies. In addition to these synthetic datasets, we apply our models to simulated single-cell RNA sequencing data generated using the SERGIO tool (Dibaeinia & Sinha, 2020), utilizing the code provided by Lorch et al. (2022) (MIT License, v1.0.5). We also test our method on neural network functional data following the setup of Brouillard et al. (2020), using the implementation from Nazaret et al. (2023) (MIT License, v0.1.0). To assess the impact of interventions, we vary the ratio of intervened variables in the set $25\%, 50\%, 75\%, 100\%$. We conduct experiments on 10 simulated datasets for each domain and each ratio of intervened variables. The observational datasets contain 5,000 samples, and each intervention dataset comprises 100 samples, mirroring the sample sizes typically found in real single-cell transcriptomics studies (Replogle et al., 2022).

We evaluate our regularized causal discovery method on synthetic datasets from four domains: linear structural equation models (SEMs), gene regulatory networks (GRNs), random Fourier features (RFFs), and neural networks (NNs). For each model type, we consider variable sizes of 10, 30, and 100 to assess scalability and performance across different problem dimensions. We use two evaluation metrics: Structural Hamming Distance (SHD) (Tsamardinos et al., 2006) and Structural Intervention Distance (SID) (Peters & Bühlmann, 2015) to compare inferred graphs to the true causal graphs. We compare to two baselines, namely GIES (Hauser & Bühlmann, 2012) and DCDI (Brouillard et al., 2020). We note that those two baselines do not scale to 100 variables. For our model, we compare the performance of our proposed causal discovery model with and without the DiffIntersort constraint (i.e. $\lambda = 0$). For the DiffIntersort score, we use the same parameters as in Chevalley et al. (2025c): $\epsilon = 0.3$ for linear, RFF and NN data, and $\epsilon = 0.5$ for GRN data, and $c = 1.0$. We use the Wasserstein distance (Villani et al., 2009) for the statistical metric. For the regularized model, we use a high value of $\lambda = 100.0$, as we do not observe a negative effect of over-regularizing, and we thus ensure that the learn potential is close to the optimal of the DiffIntersort score (see Appendix J.2 for an analysis). We present the results for the SHD metrics at 30 variables in Figure 2. The results for 10 and 100 variables for SHD can be found in the appendix in Figure 13. The results for SID can be found in Figure 15 in the appendix. Similar results for a scale-free Barabasi-Albert graph distribution can be found in the Appendix (Figures 14 and 16).

As can be seen, the DiffIntersort constraint is consistently beneficial in terms of performance on both metrics, for all types of data and at all considered scales. This comparison validates the usefulness of inducing the interventional faithfulness inductive bias to a causal models via the DiffIntersort score. It also enforces generalizability across data settings. We expect that this approach may be applicable to other causal tasks of interest, in settings where a large set of single variable interventions are available. Compared to baselines, our model outperforms on the GRN and RFF data. GIES is the best model on linear data, and DCDI has a slightly better performance on NN data. GIES and DCDI do not scale to 100 variables but we would expect the results to be the same, as our algorithm has an F1 score that is almost unaffected by the number of variables (see Figure 10 in the Appendix). The results on the F1 score also shows the robustness of our causal discovery model with the DiffIntersort constraint to the number of variables.

We also provide a proof-of-concept evaluation on the real-world protein-signaling dataset of Sachs et al. (2005), where DiffIntersort regularization improves recovery of the consensus network (see Appendix J.6 for details).

## 4    LIMITATIONS

DiffIntersort optimizes a non-convex objective, so global convergence cannot be guaranteed; to address this, we provide theoretical analysis of its optimization properties and show empirically that SORTRANKING initialization yields stable outcomes. The framework assumes acyclicity, which excludes feedback-rich systems but enables tractable theoretical guarantees. While DiffIntersort scales efficiently to thousands of variables, it does not yet reach truly massive scales, and it requires either many single-variable interventions or equivalent prior knowledge, which may not be feasible in

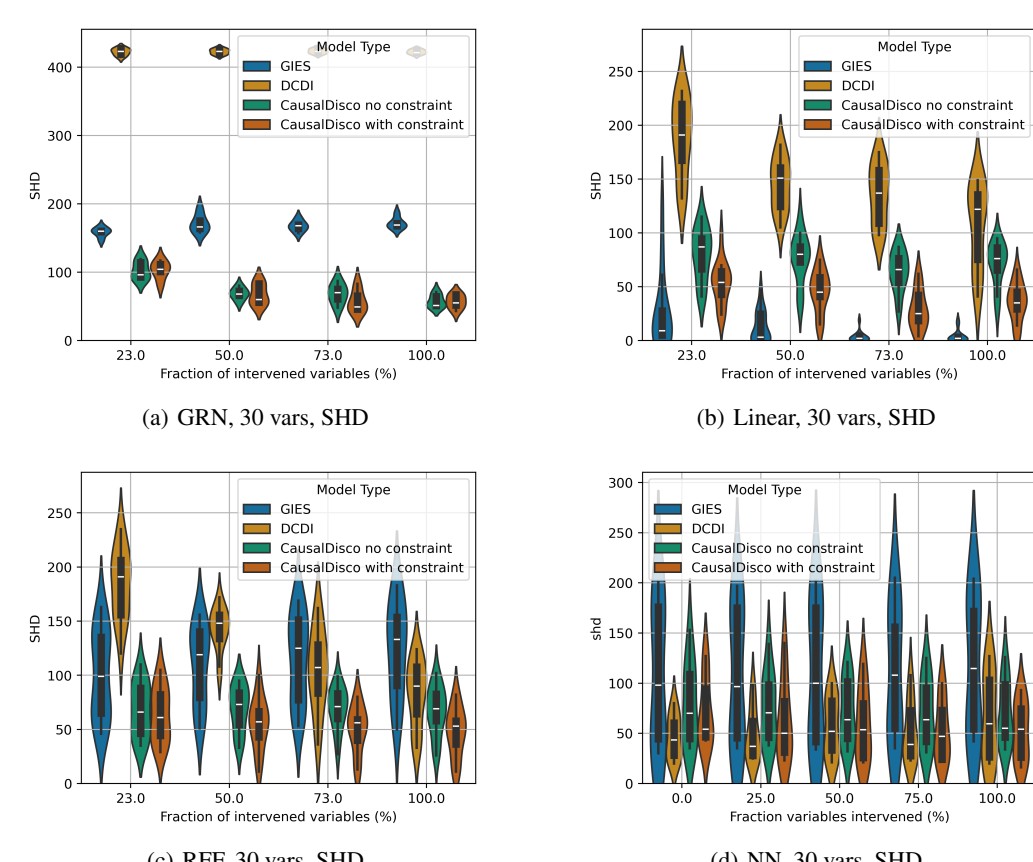

Figure 2: Comparison of SHD (lower is better) for GRN, Linear, RFF, and Neural Network data with 30 variables. Our method (CausalDisco with and without constraint) achieves lower SHD values compared to baseline methods on GRN and RFF data. GIES outperforms on the linear data and DCDI performs slightly better on NN data.

all application domains. Finally, our evaluation relies primarily on synthetic benchmarks, though we complement these with a proof-of-concept real-world experiment on protein-signaling data, which demonstrates practical utility but also highlights avenues for future work in applying to complex large-scale real-world systems.

## 5 CONCLUSION

We addressed the scalability and differentiability limitations of Intersort, a score-based method for discovering causal orderings from interventional data. By reformulating the Intersort score with differentiable sorting techniques—particularly the Sinkhorn operator—we enabled scalable and gradient-compatible optimization of causal orderings. This reformulation allows the Intersort score to serve as a continuous regularizer in gradient-based learning frameworks, enabling the flexible integration of *prior knowledge* about causal order, for example coming from interventions, expert domain knowledge (including LLMs (Vashishtha et al., 2025; Ban et al., 2023; Li et al., 2024a;b)), heuristics or external models.

Our approach preserves Intersort's theoretical strengths while substantially improving its practicality in large-scale settings. Empirical evaluations show that integrating the differentiable Intersort score into causal discovery improves performance over unregularized methods, especially in complex, nonlinear regimes with many variables. The method remains robust across data distributions and noise levels and scales effectively without loss of performance.

Beyond causal discovery, our work contributes to a broader vision: integrating causal reasoning into modern machine learning pipelines. Differentiable causal order regularization provides a principled

way to inject structured prior knowledge into learning systems. This can enhance generalization, robustness to interventions, and interpretability. In genomics, it may help enforce known regulatory hierarchies and reduce spurious associations. In reinforcement learning, it could constrain policies to respect causal dependencies, improving sample efficiency. In interpretability research, it may align model explanations with true causal influence.

More broadly, this work opens a promising direction: how can causal ordering serve as a foundation for more causally-aware deep learning models? By bridging interventional faithfulness with gradient-based learning, we move beyond purely associational representations toward models that reflect underlying causal processes. Future work may explore real-world applications, deeper theoretical analysis, and neural architectures that natively integrate causal ordering constraints—paving the way for scalable, causally grounded machine learning.

## 6 REPRODUCIBILITY STATEMENT

Complete proofs for the theoretical results are provided in Appendix G. For the simulations, the parameters used are described in Section 3, with detailed presentation of the data generation in Appendix B. The parameters of the algorithm are described in Appendix I. The corresponding code is provided as supplementary material.

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

## A  DIFFINTERSORT RELAXATION VIA SINKHORN APPROXIMATION

Inspired by Annadani et al. (2023) we define $\boldsymbol{L} \in \{0,1\}^{d \times d}$ as a matrix with upper triangular part to be 1, and vector $\boldsymbol{o} = [1, \ldots, d]^\top$. They propose the formulation

$$\text{Step}(\text{grad}(\boldsymbol{p})) = \boldsymbol{\sigma}(\boldsymbol{p})\boldsymbol{L}\boldsymbol{\sigma}(\boldsymbol{p})^\top. \tag{8}$$

and $\boldsymbol{\sigma}(\boldsymbol{p})$ is equivalent to the following optimization problem

$$\boldsymbol{\sigma}(\boldsymbol{p}) = \underset{\boldsymbol{\sigma}' \in \boldsymbol{\Sigma}_d}{\arg\max}\, \boldsymbol{p}^\top (\boldsymbol{\sigma}' \boldsymbol{o}) \tag{9}$$

where $\boldsymbol{\Sigma}_d$ represents the space of all $d$ dimensional permutation matrices. The reformulation of the permutation as an optimization problem over the set $\boldsymbol{\Sigma}_d$ can be further rewritten as

$$\boldsymbol{\sigma} = \underset{\boldsymbol{\sigma}' \in \Sigma_d}{\arg\max}\, \langle \boldsymbol{\sigma}', \boldsymbol{M} \rangle_F \tag{10}$$

where $M = \boldsymbol{p}\boldsymbol{o}^T$. Mena et al. (2018) demonstrated that this non-differentiable $\arg\max$ problem can be reformulated by regularizing it with the entropy and solving this smooth problem with the

Sinkhorn algorithm. Specifically, they showed that $\mathcal{S}(\boldsymbol{M}/t) = \arg\max_{\boldsymbol{\sigma}' \in \mathcal{B}_d} \langle \boldsymbol{\sigma}', \boldsymbol{M} \rangle + tH(\boldsymbol{\sigma}')$, where $H(\cdot)$ denotes the entropy function and the parameter $t$ controls the smoothness of the approximation, and $\mathcal{S}(\boldsymbol{M})$ is the Sinkhorn operator. The Sinkhorn operator on a matrix $\boldsymbol{M}$ involves a sequence of alternating row and column normalizations, known as Sinkhorn iterations. We refer readers to the original paper (Sinkhorn, 1964) and further applications (Adams & Zemel, 2011) of Sinkhorn operator for detailed presentation. Furthermore, we have that the regularized solution converges to the solution of Equation (10) as $t \to 0$, shown by $\lim_{t \to 0} \mathcal{S}(\boldsymbol{M}/t)$. We note here that other differentiable approximation for the permutation matrix could be used. See Appendix C.5 for a review of the differentiable sorting and ranking literature.

In practice, we approximate the limit with a value of $t > 0$ and a certain number of iterations $T$, which results in a differentiable and doubly stochastic matrix in the $d$-dimensional Birkhoff polytope $\mathcal{B}_d$, the convex hull of "hard" permutation matrices. In our experiments, we use $t = 0.05$ and $T = 500$. After applying the Sinkhorn operator to obtain a differentiable approximation of the permutation matrix, we use the Hungarian algorithm Kuhn (1955) to project it back to a valid binary permutation, ensuring consistency with the discrete causal ordering while maintaining differentiability through the straight-through estimator (Bengio et al., 2013). The resulting binary matrix is denoted as $\mathcal{S}_{\text{bin}}^T(\boldsymbol{po}^T/t)$ with "bin" emphasizing a binary-valued matrix. As a result, the score becomes differentiable and can be differentiated through the iterations of the Sinkhorn operator. By replacing the non-differentiable part of Equation (2) with this matrix, the complete form of the differentiable score (we call it DiffIntersort) is derived as

$$S(\boldsymbol{p}, \epsilon, D, \mathcal{I}, P_X^{\mathcal{C};(\emptyset)}, \mathcal{P}_{int}, t, T) = \left\langle \mathbf{D}, \left( \mathcal{S}_{\text{bin}}^T \left( \frac{\boldsymbol{po}^\top}{t} \right) \boldsymbol{L} \mathcal{S}_{\text{bin}}^T \left( \frac{\boldsymbol{po}^\top}{t} \right)^\top \right) \right\rangle_F. \tag{11}$$

In the paper, we drop the subscript "bin" and use $S(\boldsymbol{p})$ for conciseness. The maximizers of the DiffIntersort score and the Intersort score are equal for $t \to 0$ and $T \to \infty$ (Theorem 2.5). The DiffIntersort score $S(\mathbf{p})$ can be maximized with respect to the potential vector $\mathbf{p}$ using gradient descent algorithms. This allows us to find the ordering of the variables that is best aligned with the interventional data, according to the statistical distances captured in $\mathbf{D}$.

# B    DETAILED DESCRIPTION OF THE CAUSAL DISCOVERY LOSS

## B.1    LINEAR SEM PARAMETERIZATION

For our evaluation of DiffIntersort as a causal regularizer (see Section 2.3), we adopt a simple linear structural equation model (SEM). Each variable $X_j$ is generated as

$$X_j = \sum_{i=1}^{d} W_{ji} X_i + b_j + N_j, \tag{12}$$

where $\mathbf{W} \in \mathbb{R}^{d \times d}$ is the weight matrix, $b_j$ is a bias term, and $N_j$ is a noise term.

**Order masking.**    To enforce the causal ordering $\mathbf{p}$, we use the permuted upper-triangular mask

$$\boldsymbol{M}_{\mathbf{p}} = \mathcal{S}_{\text{bin}}^T(\boldsymbol{po}^\top/t) \, \boldsymbol{L} \, \mathcal{S}_{\text{bin}}^T(\boldsymbol{po}^\top/t)^\top,$$

where $\boldsymbol{L}$ is a fixed upper-triangular matrix, and $\mathcal{S}_{\text{bin}}^T$ denotes the binarized Sinkhorn operator after $T$ iterations. The effective weight matrix is then

$$\tilde{\mathbf{W}} = \mathbf{W} \circ \boldsymbol{M}_{\mathbf{p}}^T,$$

so that each variable $X_j$ may only depend on its predecessors in the order $\mathbf{p}$.

**Intervention masking.**    When evaluating interventional environments, we exclude the directly intervened variable from its own prediction loss. This ensures that the fitting objective respects the structural semantics of interventions, where incoming edges into the intervened variable are removed.

---

**Algorithm 1** DiffIntersort Causal Discovery Algorithm with Intervention Masking

1: **for** epoch $\leftarrow 1$ **to** max_epochs **do**
2:      Initialize DiffIntersort regularizer: $\mathcal{L}_{\text{DiffIntersort}} \leftarrow \lambda_2 S(\mathbf{p})$
3:      **for** each mini-batch $(\mathbf{X}_{\text{batch}}, \text{interventions}_{\text{batch}})$ **do**
4:          **Forward Pass**: $\hat{\mathbf{X}} \leftarrow f(\mathbf{X}_{\text{batch}}; \theta, \mathbf{p})$
5:          **Compute Fitting Loss $\mathcal{L}_{\text{fit}}(\theta, \mathbf{p})$:**
6:          Compute MAE for observational data:
7:          $\mathcal{L}_{\text{obs}} \leftarrow \frac{1}{n^0} \sum_{i=1}^{n^0} \ell(\mathbf{x}_i, \hat{\mathbf{x}}_i)$
8:          For each interventional environment $e \in \mathcal{E}$ with target variable $t_e$:
9:            Apply mask $m^e$ that excludes target $t_e$ from the loss
10:         $\mathcal{L}_{\text{int}}^e \leftarrow \frac{1}{n^e} \sum_{i=1}^{n^e} \ell^e\big((\mathbf{x}_i)_{\neg t_e}, (\hat{\mathbf{x}}_i)_{\neg t_e}\big)$
11:         Aggregate interventional losses relative to observational baseline:
12:         $\mathcal{L}_{\text{int}} \leftarrow \gamma \sum_{e \in \mathcal{E}} \omega^e \left( \mathcal{L}_{\text{int}}^e - \mathcal{L}_{\text{obs}} \right)$
13:         Total fitting loss: $\mathcal{L}_{\text{fit}} \leftarrow \mathcal{L}_{\text{obs}} + \mathcal{L}_{\text{int}}$
14:         **Compute Regularization Loss**:
15:         $\mathcal{L}_{\text{L1}} \leftarrow \lambda_1 \|\mathbf{W}\|_1$
16:         **Total Loss**:
17:         $\mathcal{L} \leftarrow \mathcal{L}_{\text{fit}} + \mathcal{L}_{\text{L1}}$ *Revision* $: -\mathcal{L}_{\text{DiffIntersort}}$
18:         **Backward Pass**: Compute $\nabla_{\theta, \mathbf{p}} \mathcal{L}$
19:         **Update Parameters**: $\theta, \mathbf{p} \leftarrow \text{Optimizer}(\theta, \mathbf{p})$
20:      **end for**
21: **end for**
22: **Return** Estimated causal graph and variable ordering

---

## B.2 LOSS FUNCTION

Inspired by the fitting loss in Shen et al. (2023), we define the fitting loss $\mathcal{L}_{\text{fit}}(\theta, \mathbf{p})$ as:

$$\mathcal{L}_{\text{fit}}(\theta, \mathbf{p}) = \frac{1}{n^0} \sum_{i=1}^{n^0} \ell(\mathbf{x}_i, \hat{\mathbf{x}}_i; \theta, \mathbf{p}) + \gamma \sum_{e \in \mathcal{E}} \omega^e \left( \frac{1}{n^e} \sum_{i=1}^{n^e} \ell^e\big((\mathbf{x}_i)_{\neg t_e}, (\hat{\mathbf{x}}_i)_{\neg t_e}; \theta, \mathbf{p}\big) - \frac{1}{n^0} \sum_{i=1}^{n^0} \ell^0(\mathbf{x}_i, \hat{\mathbf{x}}_i; \theta, \mathbf{p}) \right),$$
(13)

where $\ell(\cdot)$ is the mean absolute error (MAE). Each environment $e \in \mathcal{E}$ corresponds to an intervention on one variable $t_e$. To ensure that a variable does not contribute to its own prediction error under an intervention, we apply a *mask*: $(\mathbf{x}_i)_{\neg t_e}$ and $(\hat{\mathbf{x}}_i)_{\neg t_e}$ denote the observed and predicted feature vectors with the intervened target variable $t_e$ removed. $\gamma \geq 0$ balances observational and interventional terms, $\omega^e = \frac{1}{|\mathcal{E}|}$ are environment weights, and $n^e$ the number of samples in environment $e$. Environment 0 is the observational baseline. This loss encourages good fit on observational data while penalizing deviations across interventions, thereby promoting robustness consistent with causal invariance (Meinshausen, 2018).

Combining the fitting loss with regularization terms, the final training objective is:

$$\begin{aligned} \mathcal{L}(\theta, \mathbf{p}) = &\frac{1}{n^0} \sum_{i=1}^{n^0} \ell(\mathbf{x}_i, \hat{\mathbf{x}}_i; \theta, \mathbf{p}) \\ &+ \gamma \sum_{e \in \mathcal{E}} \omega^e \left( \frac{1}{n^e} \sum_{i=1}^{n^e} \ell^e\big((\mathbf{x}_i)_{\neg t_e}, (\hat{\mathbf{x}}_i)_{\neg t_e}; \theta, \mathbf{p}\big) - \frac{1}{n^0} \sum_{i=1}^{n^0} \ell^0(\mathbf{x}_i, \hat{\mathbf{x}}_i; \theta, \mathbf{p}) \right) \\ &+ \lambda_1 \|\mathbf{W}\|_1 \; Revision : -\lambda_2 S(\mathbf{p}). \end{aligned}$$
(14)

This loss consists of four components: (1) *Data fitting loss*: predictive accuracy on observational and interventional data, with intervened variables masked out; (2) *Environment invariance penalty*:

encourages consistent performance across environments; (3) $L_1$ *regularization*: promotes sparsity in $\mathbf{W}$; (4) *DiffIntersort regularization*: enforces interventional faithfulness through the score $S(\mathbf{p})$ Equation (11).

No additional acyclicity constraint is required, since acyclicity is guaranteed by masking based on the causal order $\mathbf{p}$. The full training procedure is detailed in Algorithm 1.

## C  RELATED WORK

### C.1  CAUSAL DISCOVERY METHODS

Causal discovery aims to identify cause-and-effect relationships among variables, typically represented as Directed Acyclic Graphs (DAGs). Various methodologies have been developed to infer these structures from data, primarily categorized into constraint-based, score-based, and hybrid approaches.

#### C.1.1  CONSTRAINT-BASED METHODS

These methods rely on statistical tests to assess conditional independencies in the data. The PC algorithm (Spirtes et al., 2000) is a prominent example that iteratively removes edges between variables based on conditional independence tests, constructing a skeleton of the causal graph and then orienting the edges to form a DAG. Its extension, the Fast Causal Inference (FCI) (Spirtes, 2001) algorithm, accounts for latent confounders and selection bias, providing a more robust framework in complex scenario.

#### C.1.2  SCORE-BASED METHODS

These approaches assign scores to different graph structures based on how well they fit the data and search for the graph with the optimal score. The Greedy Equivalence Search (GES) (Chickering, 2002) algorithm begins with an empty graph and incrementally adds edges to maximize a chosen score, such as the Bayesian Information Criterion (BIC). The Greedy Interventional Equivalence Search (GIES) (Hauser & Bühlmann, 2012) extends GES by incorporating interventional data, enhancing its ability to uncover causal directions that are indistinguishable using observational data alone.

#### C.1.3  FUNCTIONAL CAUSAL MODEL-BASED METHODS

These methods assume specific functional relationships between variables. For instance, the Linear Non-Gaussian Acyclic Model (LiNGAM) (Shimizu et al., 2006) assumes that the data-generating process is linear with non-Gaussian noise, enabling the identification of causal directions that are not identifiable under Gaussian assumptions.

### C.2  DIFFERENTIABLE CAUSAL DISCOVERY METHODS

Differentiable causal discovery methods have gained prominence due to their ability to integrate seamlessly with gradient-based optimization frameworks. A notable example is the NOTEARS (Zheng et al., 2018; 2020) algorithm, which formulates the structure learning problem as a continuous optimization task. It introduces a smooth characterization of the acyclicity constraint, enabling the use of standard numerical optimization techniques. However, enforcing this acyclicity constraint can be computationally intensive, especially for large-scale problems.

Building upon NOTEARS, several methods have been proposed to improve efficiency and scalability. For instance, DAGs with No Fears (Wei et al., 2020) re-examines the continuous optimization framework, addressing limitations in the original formulation and proposing enhancements to the optimization process. Similarly, DAG-NoCurl (Yu et al., 2021) introduces a no-curl constraint to ensure acyclicity, offering an alternative approach to the acyclicity enforcement in NOTEARS. Additionally, the Differentiable Causal Discovery from Interventional Data (DCDI) (Brouillard et al., 2020) method leverages interventional data to enhance identifiability and employs neural networks to model complex causal relationships. Several other prominent differentiable methods have been proposed in this line of research, including Differentiable Causal Discovery from Interventional Data

(DCDI) Brouillard et al. (2020), Stable Differentiable Causal Discovery (SDCD) (Nazaret et al., 2023), Differentiable Causal Discovery Under Latent Interventions Faria et al. (2022), Differentiable Causal Discovery with Residual Independence (DARING) He et al. (2021), and Dagma-DCE Waxman et al. (2024).

Despite these advancements, enforcing acyclicity constraints remains a challenge, often leading to increased computational complexity and potential convergence issues.

## C.3 Permutation-Based Methods

To address the challenges associated with acyclicity constraints, permutation-based methods have been developed, focusing on learning over the topological ordering of the variables. By optimizing over the permutahedron—the convex hull of all permutation vectors—these methods inherently ensure acyclicity without the need for explicit constraints.

Key developments include:

- **Greedy Sparsest Permutation (GSP)**: This method associates a score to each permutation of variables and performs a greedy search to find the permutation that leads to the sparsest DAG, effectively learning the causal structure by identifying the optimal variable ordering Solus et al. (2021).

- **Permutation-Based Causal Inference with Interventions**: Extending GSP, IGSP (Wang et al., 2017; Yang et al., 2018; Squires et al., 2020) incorporates interventional data into the permutation-based framework, enhancing the identifiability of causal structures by leveraging additional experimental information.

- **DAG Learning on the Permutahedron**: This method formulates DAG learning as an optimization problem over the permutahedron, guaranteeing the learning of a valid DAG and allowing for end-to-end training without preprocessing steps Zantedeschi et al. (2023).

- **COSMO**: Massidda et al. (2024) introduced COSMO, a constraint-free continuous optimization scheme for acyclic structure learning. At its core, COSMO employs a novel differentiable approximation of an orientation matrix parameterized by a single priority vector, enabling the learning of a smooth orientation matrix and the resulting acyclic adjacency matrix without explicitly evaluating acyclicity at any step. This approach ensures convergence to an acyclic solution and offers improved scalability due to its asymptotically faster computations.

- **QWO**: Shahverdikondori et al. (2024) introduced a novel method to enhance the efficiency of computing the optimal DAG for a given permutation, significantly speeding up permutation-based causal discovery in Linear Gaussian Acyclic Models.

- **BayesDAG**: Annadani et al. (2023) introduced BayesDAG, a framework that employs gradient-based posterior inference for causal discovery. This method utilizes stochastic gradient Markov Chain Monte Carlo (SG-MCMC) and variational inference to sample from the posterior distribution of DAGs, allowing for uncertainty quantification in the inferred causal structures. BayesDAG is applicable to both linear and nonlinear causal models, providing flexibility in modeling complex data-generating processes.

- **DP-DAG**: DP-DAG (Charpentier et al., 2022) is a differentiable probabilistic model designed for efficient DAG sampling suitable for continuous optimization. The method samples a DAG by first determining a linear ordering of nodes and then sampling edges consistent with this ordering. This approach ensures the generation of valid DAGs throughout the training process and eliminates the need for complex augmented Lagrangian optimization schemes. Additionally, the authors propose VI-DP-DAG, which combines DP-DAG with variational inference to approximate the posterior probability over DAG edges given observed data.

In summary, while differentiable causal discovery methods like NOTEARS have advanced the field by enabling continuous optimization, permutation-based methods provide a compelling alternative by focusing on learning variable orderings. This approach inherently satisfies acyclicity, offering advantages in efficiency and scalability. This approach has benefited from advances in differentiable ranking and sorting, allowing continous and differentiable relaxations of causal discovery over the permutahedron.

### C.4 CAUSAL ORDERING DISCOVERY

Causal ordering, which involves finding the causal order of the variables, is a foundational step in causal discovery. Even though it does not identify the exact graph, it can facilitate subsequent edge recovery using techniques like penalized regression (Bühlmann et al., 2014; Shimizu et al., 2011). Moreover, even without full causal graph identification, a valid causal order allows for the construction of a fully connected graph that accurately describes interventional distributions (Peters & Bühlmann, 2015; Bühlmann et al., 2014).

Recent studies have highlighted that sorting variables by variance can recover causal order in simulated datasets (Reisach et al., 2021). Building on this insight, several algorithms have been developed to infer causal order from observational data, employing methods such as score matching (Rolland et al., 2022; Montagna et al., 2023a;b).

In the context of interventional data, Tian & Pearl (2001) proposed a rule-based algorithm to infer causal order. Intersort improved on this idea by introducing a score-based method to derive the causal order, which leverages optimization tools for enhanced scalability. Additionally, Intersort provides theoretical results that upper-bound the expected error of the algorithm, particularly in scenarios where only a subset of variables is intervened upon.

### C.5 DIFFERENTIABLE SORTING AND RANKING

Differentiable sorting and ranking techniques have emerged as essential tools for integrating ranking and sorting operations into end-to-end learning frameworks. Traditional sorting operators, being non-differentiable, posed significant challenges in gradient-based optimization. To address this, Grover et al. (2019) introduced NeuralSort, a continuous relaxation of the sorting operator, enabling differentiable approximations of permutation matrices. Cuturi et al. (2019) further advanced this field by framing ranking and sorting as optimal transport problems, employing entropic regularization and Sinkhorn iterations to approximate ranks and sorted values.

Subsequent works have explored improvements in efficiency and applicability. Prillo & Eisenschlos (2020) proposed SoftSort, a simple yet effective continuous relaxation of the argsort operator, offering state-of-the-art performance with computational efficiency. Blondel et al. (2020) introduced fast differentiable sorting and ranking operators with $\mathcal{O}(n \log n)$ complexity, achieved by projecting inputs onto the permutahedron and employing isotonic optimization.

Extensions have also focused on stability and scalability. Petersen et al. (2021) presented differentiable sorting networks by relaxing conditional swap operations, addressing challenges such as vanishing gradients in large datasets. Building on this, Petersen et al. (2022) proposed monotonic differentiable sorting networks, introducing sigmoid-based relaxations to ensure gradient correctness and robustness.

### C.6 INCORPORATING PRIOR KNOWLEDGE IN CAUSAL DISCOVERY

Integrating prior knowledge is a long-standing strategy in causal discovery, especially when data alone are insufficient to identify the true graph. The form of the prior—whether edge-level, ancestral, or order-based—shapes how it can be incorporated.

**Constraint-Based Methods** Constraint-based algorithms like PC and FCI support prior knowledge as *hard constraints*—e.g., required or forbidden edges, ancestor relations, or known ordering—used to restrict or prune the search space (Meek, 1995; Borboudakis & Tsamardinos, 2012; Perković et al., 2017). These methods are symbolic and typically non-differentiable.

**Score-Based and Hybrid Approaches** Score-based and hybrid methods permit hard constraints (e.g., fixed parent sets) and soft priors (e.g., penalties on edge inclusion) (Ban et al., 2023; Li et al., 2024b; Rittel & Tschiatschek, 2023). These soft constraints are often implemented as regularization terms, with weights reflecting confidence in the prior. However, most existing work focuses on edge-level priors rather than global structural constraints like causal order.

**Differentiable Integration** Recent methods have enabled differentiable incorporation of edge-level prior knowledge within continuous optimization frameworks. For example, Chowdhury et al. (2023)

add linear constraints to NOTEARS, while Li et al. (2024a) propose KEEL, which uses fuzzy soft penalties for expert priors in a differentiable DAG constraint. In probabilistic models, Rittel & Tschiatschek (2023) introduce DPM-DAG, which encodes edge priors as a differentiable prior over DAGs.

While these approaches support end-to-end training, they primarily operate at the level of individual edges. In contrast, our work introduces a differentiable score over causal orderings, enabling the integration of global structural prior knowledge—e.g., partial or full variable orderings—into gradient-based learning. This formulation naturally captures priors derived from interventional data, domain heuristics, or proxy models. Furthermore, our method scales efficiently and can act as a regularizer in downstream tasks, bridging structured causal priors with deep learning.

Revision:

# D    THEOREMS FROM CHEVALLEY ET AL. (2025C)

We recall the statement of the two theorems from Chevalley et al. (2025c) used as theoretical upper-bounds. Theorem D.1 is Theorem 2 in the original paper and Theorem D.2 corresponds to Theorem 4.

**Theorem D.1.** *Assume that we are given $P_X^{\mathcal{C},(\emptyset)}$ and $P_X^{\mathcal{C},do(X_k:=\tilde{N}_k)}, \forall k \in \mathcal{I}$, such that $(\tilde{N}, \mathcal{C})$ is $\epsilon$-interventionally faithful for some $\epsilon > 0$, and let $\mathcal{I}$ be chosen uniformly at random, where $p_{int} := P(i \in \mathcal{I}) \forall i \in V, 0 < p_{int} < 1$, then $\mathbb{E}[D_{top}(\mathcal{G}, \pi_{opt})] \leq \sum_{(i,j) \in \mathcal{G}} (1 - p_{int})^{|AN_j^{\mathcal{G}} \cup \{j\} \setminus AN_i^{\mathcal{G}}|}$.*

**Theorem D.2.** *Let $\mathcal{I}$ be chosen uniformly at random, $\mathcal{G}$ be a random Erdos-Renyi directed acyclic graph with edge probability $p_e$, where $p_{int} := P(i \in \mathcal{I}) \forall i \in V, 0 < p_{int}$, then $\mathbb{E}[D_{top}(\mathcal{G}, \pi_{opt})] \leq \frac{(1-p_{int})^2}{p_{int}} \left[ d - (1 - p_{int} p_e) \frac{1-(1-p_{int} p_e)^d}{p_{int} p_e} \right]$. There is also a looser, but independent of $p_e$, bound $\mathbb{E}[D_{top}(\mathcal{G}, \pi_{opt})] \leq \frac{(1-p_{int})^2}{p_{int}} d$.*

# E    INTERSORT OPTIMIZATION DETAILS

Chevalley et al. (2025c) propose the Intersort algorithm to optimize the score. Specifically, the Intersort algorithm consists of two steps. The first step, SORTRANKING, finds an initial by ordering the observed statistical $D\left(P_{X_j}^{\mathcal{C},(\emptyset)}, P_{X_j}^{\mathcal{C},do(X_i:=\tilde{N}_i)}\right)$ distances from highest to lowest, adding an edge to the solution if it does not create a cycle. When the significance threshold $\epsilon$ is reached, the algorithm stops and returns the topological order of the built graph as an initial solution for the second step. This runtime complexity of this algorithm is $\mathcal{O}(d \cdot |\mathcal{I}| \log(d \cdot |\mathcal{I}|))$. The second step, LOCALSEARCH, iteratively searches in a close neighborhood in permutation space for a higher scoring solution, until the score cannot be improved anymore. For each iteration, the complexity is $\mathcal{O}(d^2)$, and the number of iterations is approximately $\mathcal{O}(d)$.

# F    DIFFINTERSORT OPTIMIZATION PROPERTIES

The operator $S(\mathbf{M})$ of Theorem 2.6 is computed using the Sinkhorn operator and its gradient is estimated from the backward pass through the Sinkhorn iterations, and thus the Lipschitz continuity of the gradients holds for Equation (11) only when $T \to \infty$. The convergence of the forward and backward pass of the Sinkhorn approximation have been thoroughly studied, particularly in the optimal transport literature (Eisenberger et al., 2022; Greco et al., 2023; Pauwels & Vaiter, 2023). In practice, we also ensure that $\mathbf{p}$ is bounded by passing it through the Sigmoid function. The Lipschitz constant of Theorem 2.6 implies that the learning rate needs to be appropriately scaled depending on $t$ and $d$. We also note that because we apply the Hungarian algorithm and estimate the gradient with the straight-through estimator, the gradients of Equation (11) are biased compared to the gradients of the soft permutation version of the score.

Beyond the theoretical guarantees, we empirically examined the effect of the Sinkhorn parameters in Appendix J.1. The results confirm the expected trade-off: lower temperatures $t$ and larger iteration counts $T$ improve assignment fidelity but increase runtime and memory, while higher $t$ yields

smoother gradients and faster convergence at the cost of bias. In practice, we found that accuracy gains saturate quickly—particularly in $T$—and that a moderate choice of $t = 0.05$ and $T = 500$ balances stability, efficiency, and accuracy across problem scales.

# G    PROOFS

*Proof of Theorem 2.5.* First, let us recall that we have $\boldsymbol{p} \in \mathbb{R}^d$, and $\pi \in \{0, 1, \ldots, d\}^d$, where $\forall i, j \in \{0, 1, \ldots, d\}, \pi_i \neq \pi_j$. We thus trivially have that any permutation $\pi$ can be represented by a potential $\boldsymbol{p}$, by $\boldsymbol{p}_i = -\pi_i \forall i \in \{0, 1, \ldots, d\}$. We now have to prove that if $\pi \in \Pi$, then the corresponding potential $\boldsymbol{p}_\pi \in \mathbb{P}$. Let $s = \max_\pi S(\pi, \epsilon, D, \mathcal{I}, P_X^{\mathcal{C}, (\emptyset)}, \mathcal{P}_{int}, c)$ be the maximum achievable score. The sum of the score is over the elements of $\mathbf{D}_{ij}$ where $\pi_i < \pi_j$. For all these pairs of indices, we also have that $p_{\pi_i} > p_{\pi_j}$, and thus for all those pairs, we also have $(\text{Step}(\text{grad}(\boldsymbol{p}_\pi)))_{ij} = 1$. This exactly corresponds to the elements that are non-zero and thus contribute to the sum in Equation (3). Thus we have that $S(\boldsymbol{p}_\pi) = s$, and as such $\boldsymbol{p}_\pi \in \mathbb{P}$, which concludes the proof. $\qquad\square$

*Proof of Theorem 2.6.* **Step 1: Differentiation with respect to $S$.**

For $g(S) = \langle D, \ SLS^\top \rangle_F$ and any perturbation $dS$,

$$dg \ = \ \langle D, \ dS\, L\, S^\top + S\, L\, dS^\top \rangle_F \ = \ \big\langle DSL^\top + D^\top SL, \ dS \big\rangle_F,$$

hence

$$\nabla_S g(S) \ = \ D\, S\, L^\top + D^\top S\, L. \tag{15}$$

**Step 2: Chain rule from $p$ to $S(M(p))$.**

Let $S(p) := S(M(p))$ with $M(p) = p\, o^\top$. A variation $dp$ induces $dM = (dp)\, o^\top$ and

$$\nabla f(p)\, dp = \big\langle \nabla_S g\big(S(p)\big), \ \nabla S(M(p))\big[\, dM\, \big] \big\rangle_F$$
$$= \big\langle \nabla_S g\big(S(p)\big), \ \nabla S(M(p))\big[\, dp\, o^\top\, \big] \big\rangle_F.$$

Thus $\nabla f(p)$ is the composition of the linear maps $\delta p \mapsto \delta p\, o^\top$, then $\nabla S(M(p))$, then inner product with $\nabla_S g(S(p))$.

**Step 3: Regularity and uniform bounds for the Sinkhorn map.**

Consider the strictly concave program in $P$ with linear equality constraints defining $S(M)$. The KKT system is nonsingular at every $M$, and by the implicit function theorem the solution map $S$ is $C^\infty$ in a neighborhood of any $M$. On the compact set

$$\mathcal{M} \ = \ \big\{ M(p) = p\, o^\top \, : \ p \in [-1, 1]^d, \ \|o\|_\infty \leq B_o \big\}$$

we have $\|M\|_\infty \leq B_o$. Standard entropic-regularization arguments imply a uniform lower bound on the entries of $P = S(M)$:

$$\min_{i,j} S(M)_{ij} \ \geq \ \underline{p}\big(d, B_o/t\big) \ > \ 0,$$

depending only on $d$ and $B_o/t$. This yields uniform conditioning of the KKT matrix on $\mathcal{M}$ and therefore finite bounds

$$\|\nabla S(M)\|_{\text{op}} \ \leq \ A_1\, \Phi_1\Big(\tfrac{B_o}{t}\Big)\, \Psi_1(d), \qquad \text{Lip}\big(\nabla S; \mathcal{M}\big) \ \leq \ A_2\, \Phi_2\Big(\tfrac{B_o}{t}\Big)\, \Psi_2(d), \tag{16}$$

for universal $A_1, A_2 > 0$, with $\Phi_1, \Phi_2$ nondecreasing and $\Psi_1, \Psi_2$ nondecreasing.

**Step 4: Lipschitz bound for $\nabla f$.**

Let $p_1, p_2 \in [-1, 1]^d$ and write $M_i = M(p_i)$, $S_i = S(M_i)$. Using equation 15, the product rule inside the composition described in Step 2, submultiplicativity of operator norms, and the fact that a doubly stochastic matrix has $\|S_i\|_2 \leq 1$, we obtain

$$\|\nabla f(p_1) - \nabla f(p_2)\|_2 \leq \ \big\|\big(\nabla_S g(S_1) - \nabla_S g(S_2)\big) \circ \nabla S(M_1)\big\| \, \|o\|_2\, \|p_1 - p_2\|_2$$
$$+ \ \big\|\nabla_S g(S_2) \circ \big(\nabla S(M_1) - \nabla S(M_2)\big)\big\| \, \|o\|_2\, \|p_1 - p_2\|_2$$
$$\leq \ 2\, \|D\|_2\, \|L\|_2\, \|S_1 - S_2\|_2\, \|\nabla S(M_1)\|_{\text{op}}\, \|o\|_2\, \|p_1 - p_2\|_2$$
$$+ \ 2\, \|D\|_2\, \|L\|_2\, \text{Lip}\big(\nabla S; \mathcal{M}\big)\, \|M_1 - M_2\|_F\, \|o\|_2\, \|p_1 - p_2\|_2.$$

By the mean-value theorem for $S$ on $\mathcal{M}$, $\|S_1 - S_2\|_2 \leq \|\nabla S(M^\xi)\|_{\text{op}} \|M_1 - M_2\|_F$ for some $M^\xi$ on the segment, and $\|M_1 - M_2\|_F \leq \|o\|_2 \|p_1 - p_2\|_2$. Using $\|o\|_2 \leq \sqrt{d} \|o\|_\infty \leq \sqrt{d} B_o$, the bounds equation 16, and collecting monotone factors into

$$\Phi = \max\{\Phi_1^2, \Phi_2\}, \qquad \Psi = \max\{\Psi_1^2, \Psi_2\},$$

we arrive at

$$\|\nabla f(p_1) - \nabla f(p_2)\|_2 \leq C_0 B_D B_L B_o^2 \Phi\left(\tfrac{B_o}{t}\right) \Psi(d) \|p_1 - p_2\|_2,$$

for a universal $C_0 > 0$. The functions $\Phi$ and $\Psi$ are nondecreasing by construction, hence the right-hand side increases when $t$ decreases (via $B_o/t$) and when $d$ increases. This proves the claim. □

*Proof of Proposition 2.8.* Let $\boldsymbol{p}^1, \boldsymbol{p}^2 \in \boldsymbol{P}^D$, and $\boldsymbol{p}^c = \lambda \boldsymbol{p}_1 + (1-\lambda)\boldsymbol{p}_2, 0 < \lambda < 1$ be their convex combination. For $\boldsymbol{p}^c$ to also be part of $\boldsymbol{P}^D$, we must have that for all $(i,j)$ such that $\boldsymbol{D}_{ij} > 0$, we have $\boldsymbol{p}_i^c > \boldsymbol{p}_j^c$. For all such $(i,j)$, we have $\boldsymbol{p}_i^1 > \boldsymbol{p}_j^1$ and $\boldsymbol{p}_i^2 > \boldsymbol{p}_j^2$. Multiplying the first inequality by $\lambda$ and the second one by $(1-\lambda)$, and summing them together gives $\lambda \boldsymbol{p}_i^1 + (1-\lambda)\boldsymbol{p}_i^2 > \lambda \boldsymbol{p}_j^1 + (1-\lambda)\boldsymbol{p}_j^2$, which implies that $\boldsymbol{p}_i^c > \boldsymbol{p}_j^c$, concluding the proof that $\boldsymbol{P}^D$ is a convex set. □

*Proof of Theorem 2.9.* We first show that $\boldsymbol{P}^* \subseteq \boldsymbol{P}^D$. Equivalently, we can show that for all $\boldsymbol{p}^* \in \arg\max_{[-1,1]^d} \lim_{t \to 0} f(\boldsymbol{p}) = \boldsymbol{P}^*, \boldsymbol{p}_i^* > \boldsymbol{p}_j^*$ if $\boldsymbol{D}_{ij} > 0$. First, we note that there exist such a solution, given that the data is $\epsilon$-interventionally faithfull, all $\boldsymbol{D}_{ij} > 0$ do not create cyclic causal effect, and they thus can be represented by a DAG. Now we argue that if $\boldsymbol{p}_i > \boldsymbol{p}_j$ for some $\boldsymbol{D}_{ij} > 0$, then $\boldsymbol{p} \notin \boldsymbol{P}^*$. From the proof of Lemma 4 in Chevalley et al. (2025c), a necessary condition for a solution to be optimal is that all $\boldsymbol{D}_{ij} > 0$ follow the causal order, if the constant $c$ is large enough. We thus must have that $\boldsymbol{P}^* \subseteq \boldsymbol{P}^D$.

Second, we want to show that $\forall \boldsymbol{p} \in \boldsymbol{P}^D, |\lim_{t \to 0} f(\boldsymbol{p}) - \lim_{t \to 0} f(\boldsymbol{p}^*)| \leq \epsilon \frac{d(d-1)}{2}$. From the fact that both potentials are in $\boldsymbol{P}^D$, their score difference cancels out all the $\boldsymbol{D}_{ij} > 0$. As such, the score of the two potentials only differ in the other positions, where $-\epsilon \leq \boldsymbol{D}_{ij} \leq 0$. Given that there are at most $\frac{d(d-1)}{2}$ such entries, there difference is bounded by $\epsilon \frac{d(d-1)}{2}$, which concludes the proof. □

*Proof of Lemma 2.10.* By construction, SORTRANKING traverses the distances $D_{ij}$ from largest to lowest, stopping when a distance $D_{ij} \leq \epsilon$ is reached. SORTRANKING adds an edge to the solution if it does not create a cycle in the DAG. As such, the topological order of the final solution is in $\boldsymbol{P}^D$ if it can add all pairs $\{(i,j), D_{ij} > \epsilon\}$ to the solution. Given that the data is $\epsilon$-interventionally faithfull, we have that $D_{ij} > \epsilon$ only if $i \to j$ is a directed path in the true graph $\mathbf{G}$. Given that $\mathbf{G}$ is a DAG, adding $(i,j)$ to the solution of SORTRANKING cannot create a cycle, and as such all pairs $\{(i,j), D_{ij} > \epsilon\}$ are in the final solution of SORTRANKING, which implies that the toplogical order of the final solution is in $\boldsymbol{P}^D$. □

# H  DETAILS OF EMPIRICAL EVALUATION

We here describe the setting of our synthetic evaluation. We follow the setup of Chevalley et al. (2025c), which was based on the setup and implementation of Lorch et al. (2022). For the linear and RFF domains, the noise distribution is chosen uniformly at random from the following options: uniform Gaussian (noise scale independent of the parents), heteroscedastic Gaussian (noise scale functionally dependent on the parents), and Laplace distribution. In the neural network domain, the noise distribution is Gaussian with a fixed variance. All datasets are standardized based on the mean and variance of the observational data to eliminate the Varsortability artifact identified by Reisach et al. (2021).

## H.1  LINEAR AND RANDOM FOURIER FEATURE (RFF) DOMAINS

Each causal variable $x_j$ is modeled in terms of its parents $x_{\text{pa}(j)}$ using the equation:

$$x_j \leftarrow f_j(x_{\text{Pa}_j^\mathcal{G}}, \epsilon_j) = f_j(x_{\text{Pa}_j^\mathcal{G}}) + h_j(x_{\text{Pa}_j^\mathcal{G}})\epsilon_j,$$

where $\epsilon_j$ denotes additive noise, potentially heteroscedastic. The noise scale $h_j(x)$ is specified as:

$$h_j(x) = \log(1 + \exp(g_j(x))),$$

with $g_j(x)$ being a nonlinear function. For heteroscedastic noise, random Fourier features are used, configured with a length scale of 10.0 and output scale of 2.0.

Interventions fix the value of the intervened variable to a constant drawn from a signed Uniform distribution over $[1.0, 5.0]$.

### H.1.1 DOMAIN-SPECIFIC MODELING

- **Linear Domain:** Causal functions are linear transformations:

$$f_j(x_{\mathrm{Pa}_j^{\mathcal{G}}}) = w_j^\top x_{\mathrm{Pa}_j^{\mathcal{G}}} + b_j,$$

   where $w_j$ and $b_j$ are sampled independently. Specifically, $w_j$ is drawn from a signed Uniform distribution over $[1, 3]$, and $b_j$ is sampled from a Uniform distribution over $[-3, 3]$.

- **RFF Domain:** Causal functions are modeled using a Gaussian Process (GP) approximation via random Fourier features:

$$f_j(x_{\mathrm{Pa}_j^{\mathcal{G}}}) = b_j + c_j \sqrt{\frac{2}{M}} \sum_{m=1}^{M} \alpha^{(m)} \cos\left(\frac{1}{\ell_j} \omega^{(m)} \cdot x_{\mathrm{Pa}_j^{\mathcal{G}}} + \delta^{(m)}\right),$$

   where $\alpha^{(m)} \sim \mathcal{N}(0, 1)$, $\omega^{(m)} \sim \mathcal{N}(0, \mathbf{I})$, and $\delta^{(m)} \sim \mathrm{Uniform}(0, 2\pi)$. Parameters $b_j$, $c_j$, and $\ell_j$ are sampled independently: $\ell_j$ from Uniform($[7.0, 10.0]$), $c_j$ from Uniform($[10.0, 20.0]$), $b_j$ from Uniform($[-3, 3]$), and $M = 100$.

## H.2 SIMULATION OF SINGLE-CELL GENE EXPRESSION DATA

Realistic single-cell RNA sequencing data is generated using the SERGIO simulator (Dibaeinia & Sinha, 2020). SERGIO models gene expression as snapshots from the steady state of a dynamical system governed by the chemical Langevin equation. Gene interactions are defined by a causal graph $G$, with variability introduced through master regulator (MR) rates. Cell types are distinguished by differences in MR rates, which affect noise and expression profiles.

### H.2.1 SIMULATION PARAMETERS

Simulations cover $c = 5$ cell types and $d$ genes. Key parameters include:

- Interaction strengths $k$: Uniform($[1.0, 5.0]$),
- MR production rates $b$: Uniform($[1.0, 3.0]$),
- Hill coefficients: $\gamma = 2.0$,
- Decay rates: $\lambda = 0.8$,
- Noise scale: $\zeta = 1.0$.

Interventions correspond to gene knockouts, where expression is fixed at 0. Technical noise is not simulated.

## H.3 SIMULATION OF NEURAL NETWORK-BASED DATA

To simulate data for causal discovery, random fully connected neural networks (MLPs) are used to define conditional distributions.

### H.3.1 NEURAL NETWORK SPECIFICATION

Each MLP has a single hidden layer of 10 neurons and uses ReLU activation. The MLP maps inputs $x_{\mathrm{Pa}_j^{\mathcal{G}}}$ to a scalar output representing the mean $\mu$ of a conditional Gaussian:

$$p_j(x_j|x_{\mathrm{Pa}_j^{\mathcal{G}}}) \sim \mathcal{N}(\mu = \mathrm{MLP}(x_{\mathrm{Pa}_j^{\mathcal{G}}}), \sigma = 1.0).$$

### H.3.2 INTERVENTIONAL DATA GENERATION

Interventions alter the distribution of affected nodes. For an intervened node, the distribution is set to:

$$p_j(x_j|\text{do}(x_j)) \sim \mathcal{N}(2, 1.0),$$

independent of the MLP, to simulate intervention effects.

## I  HYPERPARAMETERS

Table 1: Hyperparameters for the DiffIntersort Causal Discovery Algorithm

| Parameter | Value | Description |
|---|---|---|
| $\lambda_1$ | 0.01 | L1 regularization coefficient for the weight matrix $\mathbf{W}$. |
| $\lambda_2$ | 100.0 | Regularization parameter for the DiffIntersort regularization |
| scaling $c$ | Dimension dependent | Scaling factor for the distance matrix (see Table 2). |
| n_iter | 2000 | Maximum total number of iterations for the optimization process. |
| lr_int | Dimension dependent | Learning rate for the permutation optimizer parameters (see Table 2). |
| n_iter_sinkhorn | 500 | Number of iterations for the Sinkhorn normalization process. |
| t_sinkhorn | 0.05 | Temperature parameter for the Sinkhorn normalization. |
| eps | 0.3 or 0.5 for GRN | Epsilon value for the distance matrix. |
| p_scale | 0.001 | Initial scaling factor for the initialization of the permutation vector $\mathbf{p}$. |
| Number batches | 3 | Number of mini-batches per iterations. |
| $\gamma$ | 0.5 | Parameter controlling the emphasis on invariance across environments. |
| betas | (0.9, 0.99) | Beta parameters for the Adam optimizer of the potential. |
| lr_weights | 1e-3 | Learning rate for the data fitting parameters. |

Table 2: Configuration Parameters for Different Dimensions

| Dimension | Learning Rate (lr) | Scaling $c$ |
|---|---|---|
| 3 | 0.05 | 0.1 |
| 10 | 0.05 | 0.5 |
| 30 | 0.01 | 1.0 |
| 100 | 0.001 | 1.0 |
| 1000 | 0.0005 | 1.0 |
| 2000 | 0.0001 | 1.0 |

## J  ADDITIONAL EXPERIMENTS

### J.1  SCALABILITY PROPERTIES OF DIFFINTERSORT

We next examine how DIFFINTERSORT scales in both runtime and accuracy compared to IN-TERSORT on the task of causal order discovery. We consider intervention coverage $p_{\text{int}} = 0.5$, multiple graph families (Erdős–Rényi and Barabási–Albert), and dimensionalities $d \in \{5, 10, 20, 30, 40, 50, 60, 70, 80, 90, 100, 200\}$. Results are averaged over several seeds and graph instances (see Table 3 for the graph densities used). Runtime is reported in seconds on a logarithmic scale, and accuracy is measured via top-order divergence $D_{\text{top}}$ (lower is better). We benchmark DIFFINTERSORT on both CPU and GPU backends to illustrate the effect of parallelization.

As shown in Figure 3, the runtime of INTERSORT grows steeply with $d$, while DIFFINTERSORT exhibits much slower growth and becomes faster than Intersort beyond $d \approx 60$. GPU acceleration yields further gains starting around $d = 200$, enabling applications up to several thousand variables. In terms of accuracy (Figure 4), both algorithms perform similarly at small scale, but beyond $d \approx 50$ DIFFINTERSORT consistently achieves lower $D_{\text{top}}$. This reflects the advantage of differentiable optimization: smooth gradients provide stable updates and allow coordinated improvements across many variables, whereas discrete local search is prone to stalling in suboptimal orders. Together, these results show that DIFFINTERSORT not only scales more favorably in time and memory, but also benefits from its differentiable formulation to recover more accurate orders at larger dimensions. This can be explained by the difference in optimization paradigms: gradient-based methods explore smoother landscapes and update all variables jointly, whereas discrete local search heuristics are prone to stalling in suboptimal neighborhoods. Similar distinctions have been observed in the broader optimization literature, where gradient-based refinements often outperform purely derivative-free or evolutionary heuristics (Jamieson et al., 2012; Rios & Sahinidis, 2013; Salomon, 2002). Viewed from this perspective, DIFFINTERSORT can be seen as a hybrid approach: discrete search provides an effective initialization, while gradient refinement yields better solutions as dimensionality grows.

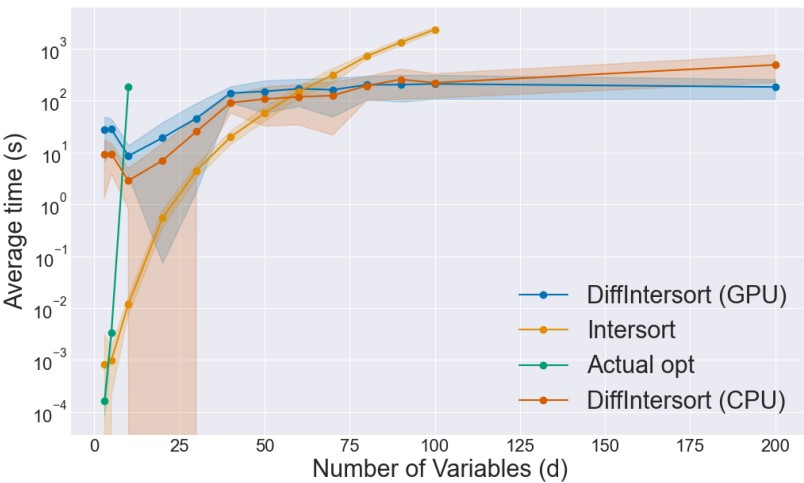

Figure 3: We evaluate the computational scalability of the tested algorithms. The plot shows the average execution time in seconds (y-axis) as the number of variables ($d$, x-axis) increases. Note that the y-axis is on a logarithmic scale to effectively display the wide variance in runtime performance between algorithms. Each line represents a specific algorithm, with DiffIntersort's performance detailed for both CPU and CUDA platforms. The solid line indicates the mean execution time averaged over all experimental settings, while the shaded area denotes the standard deviation.

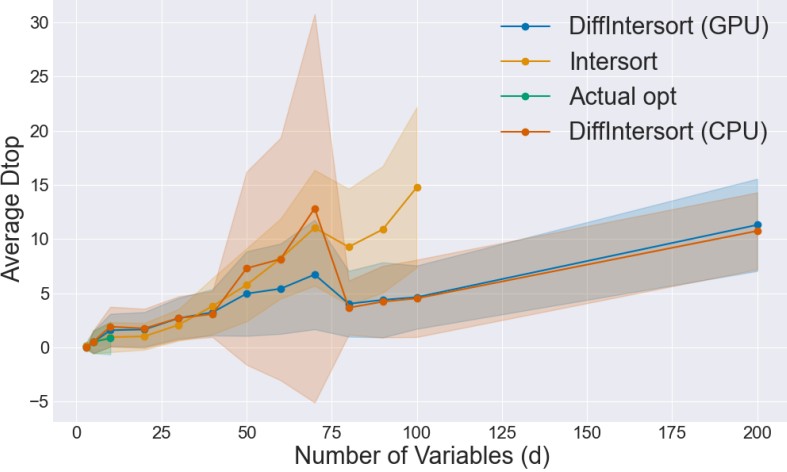

Figure 4: This figure presents a comparative analysis of algorithm accuracy as a function of problem dimensionality. The y-axis represents the average top-order divergence (Dtop), a metric where lower values signify higher accuracy in causal ordering. The x-axis indicates the number of variables ($d$) in the graph. Each line corresponds to a distinct algorithm, with the DiffIntersort method being benchmarked on both CPU and CUDA architectures. The data points represent the mean d_top value, averaged across all experimental conditions, including multiple simulation runs and graph types. The shaded region surrounding each line illustrates the standard deviation.

Table 3: Scalability experiment settings. For each number of variables $d$, we report the edge density for Erdős–Rényi (ER) graphs ($p_e$) and the average number of edges per variable for scale-free (SF) graphs. Intervention coverage was fixed to $p_{int} = 0.5$. DiffIntersort was always run on CPU and GPU, while Intersort and Actual Opt were only run upto $d = 100$ and $d = 10$ respectively.

| $d$ | ER $p_e$ values | SF avg. edges/var |
|-----|-----------------|-------------------|
| 5 | {0.5, 0.66, 0.75} | {1, 2, 3} |
| 10 | {0.5, 0.66, 0.75} | {1, 2, 3} |
| 20 | {0.05, 0.10, 0.20} | {1, 2, 3} |
| 30 | {0.05, 0.10, 0.20} | {1, 2, 3} |
| 40 | {0.05, 0.10, 0.20} | {1, 2, 3} |
| 50 | {0.05, 0.10, 0.20} | {1, 2, 3} |
| 60 | {0.05, 0.10, 0.20} | {1, 2, 3} |
| 70 | {0.05, 0.10, 0.20} | {1, 2, 3} |
| 80 | {0.02, 0.01, 0.005} | {1, 2, 3} |
| 90 | {0.02, 0.01, 0.005} | {1, 2, 3} |
| 100 | {0.02, 0.01, 0.005} | {1, 2, 3} |
| 200 | {0.02, 0.01, 0.005} | {1, 2, 3} |

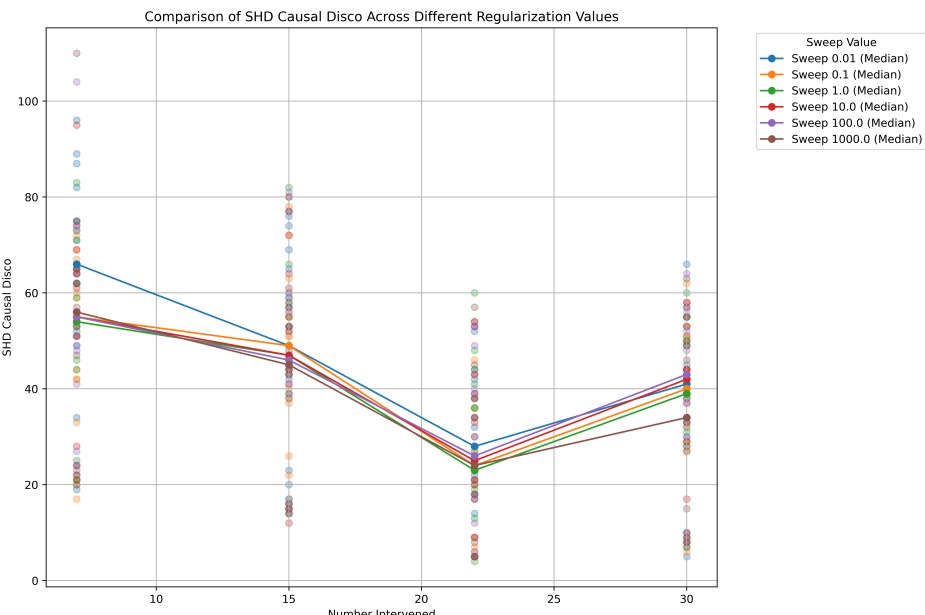

Figure 5: Comparison of SHD Causal Disco across different regularization values (Sweep) on Linear data with 30 variables. The x-axis represents the number of interventions, while the y-axis shows the Structural Hamming Distance (SHD) for causal discovery. Transparent scatter points indicate individual data samples, while solid lines connect the median SHD values at each intervention level for each sweep value. Lower SHD values indicate better causal structure recovery. The plot highlights how different regularization strengths impact performance across varying intervention numbers.

## J.2 REGULARIZATION SWEEP

We test different values for the regularization strength of DiffIntersort in causal discovery (see Figure 5). We observed that there does not seem to be major differences and that there are no risks of over-regularization. We thus use a value of $\lambda_2 = 100.0$ for all experiments.

## J.3 SIMULATED DISTANCE MATRICES

Figure 1 in the main text summarized our large-scale results at $d = 2000$ for both Erdős–Rényi (ER) and scale-free (SF) networks. To provide a more complete picture, we report here extended experiments across a wider range of graph sizes ($d \in \{5, 30, 100, 1000\}$) and densities (corre-

sponding to roughly 1, 2, or 3 edges per variable). As before, intervention coverage varies over $p_{int} \in \{0.25, 0.33, 0.5, 0.66, 0.75\}$, and for each setting we evaluate multiple random graph draws and intervention configurations.

Figures 6 and 7 show bar plots of the $D_{top}$ divergence for ER and SF graphs, respectively, comparing DiffIntersort against Intersort, SORTRANKING, and the theoretical bounds of Chevalley et al. (2025c), as well as Actual Opt for $d = 5$. Actual Opt corresponds to the brute-force algorithm that scores all possible permutations and picks the best one. These results complement Figure 1 by illustrating how performance trends evolve across scales: while all methods behave similarly at very small $d$, the advantages of DiffIntersort become more visible as dimensionality grows, particularly on SF graphs.

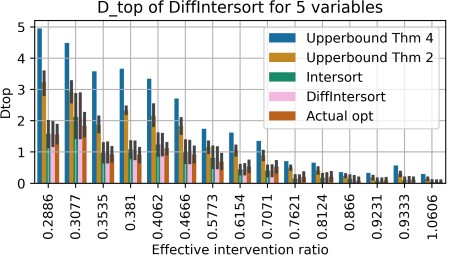

(a) Simulation ER with 5 variables

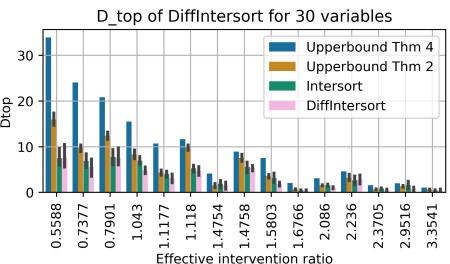

(b) Simulation ER with 30 variables

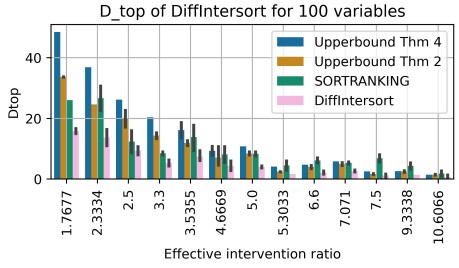

(c) Simulation ER with 100 variables

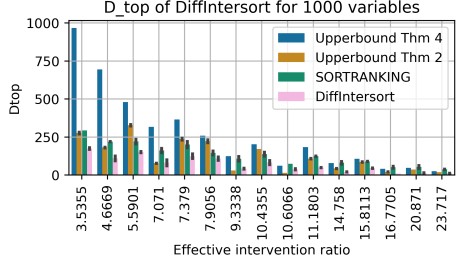

(d) Simulation ER with 1000 variables

Figure 6: Comparison of performance on simulated ER graphs in terms of $D_{top}$ divergence between the two bounds of (Chevalley et al., 2025c), DiffIntersort, Intersort, and SORTRANKING. For each setting, we draw multiple graphs, where a setting is the tuple $(p_{int}, p_e)$. Then, for each graph, we run the algorithm on multiple configurations, where a configuration corresponds to a set of intervened variables following $p_{int}$. We have $p_{int} \in \{0.25, 0.33, 0.5, 0.66, 0.75\}$ for all scales. For 5 variables, $p_e \in \{0.5, 0.66, 0.75\}$. For 30, $p_e \in \{0.05, 0.1, 0.2\}$. For 1000 variables, $p_e \in \{0.005, 0.002, 0.001\}$. These probabilities approximately correspond to an average of 1, 2, or 3 edges per variable.

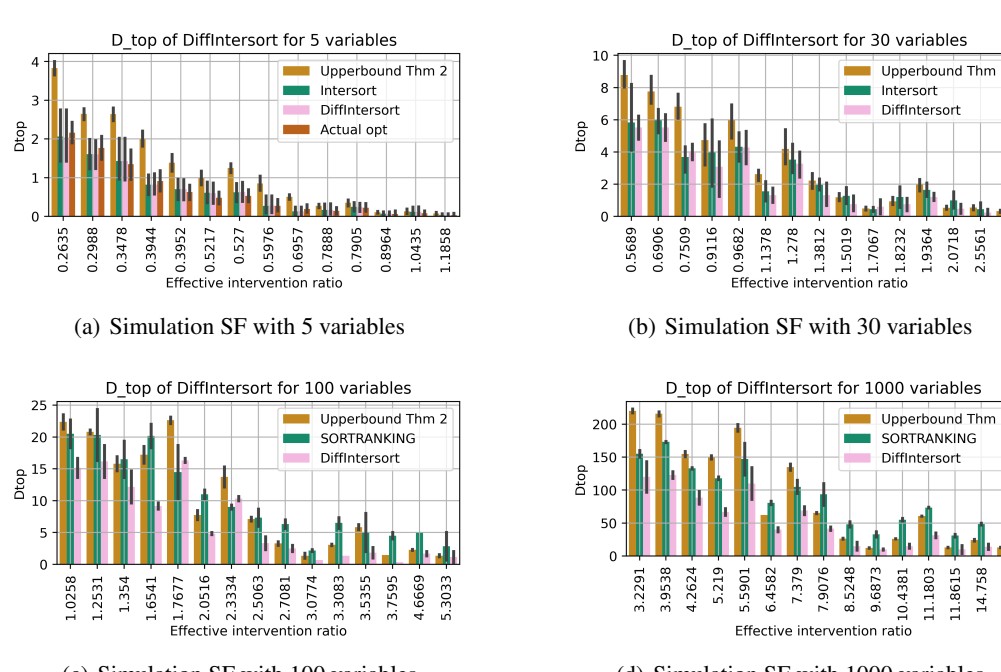

(a) Simulation SF with 5 variables

(b) Simulation SF with 30 variables

(c) Simulation SF with 100 variables

(d) Simulation SF with 1000 variables

Figure 7: Comparison of performance on simulated SF graphs in terms of $D_{top}$ divergence between the two bounds of (Chevalley et al., 2025c), DiffIntersort, Intersort and SORTRANKING. For each setting, we draw multiple graphs, where a setting is the tuple $(p_{int}, p_e)$. The networks follow a Barabasi-Albert SF distribution, with average edge per variable in $\{1, 2, 3\}$. A setting is the tuple $(p_{int}, p_e)$, where $p_e = \frac{2E(\#edges)}{d(d-1)}$. Then, for each graph, we run the algorithm on multiple configurations, where a configuration corresponds to a set of intervened variables following $p_{int}$. We have $p_{int} \in \{0.25, 0.33, 0.5, 0.66, 0.75\}$ for all scales.

## J.4 SINKHORN PARAMETERS ANALYSIS

To make our relaxation choices explicit and reproducible, we evaluate how DiffInterSort depends on the entropic Sinkhorn temperature $t$ and the number of Sinkhorn iterations $T$. These hyperparameters follow a core trade-off: smaller $t$ and larger $T$ reduce the relaxation gap and improve fidelity, but may increase runtime and memory requirements; conversely, larger $t$ or smaller $T$ speed training with smoother gradients (see Theorem 2.6) at the cost of bias. We therefore sweep ($t \in \{0.01, 0.05, 0.1, 0.5\}$, $T \in \{100, 500, 1000\}$) across problem sizes, for an intervention coverage of 50% and a scale-free graph distribution with 3 edges per variable, and report accuracy ($D_{top}$), and compute (wall-clock runtime). Results are presented in Figure 8 and Figure 9. As expected, higher $T$ and lower $t$ lead to higher compute time. However, the gain in accuracy is low, especially for the number of iterations $T$. We also observed memory issue in high dimensions ($d \geq 1000$), and we thus opt for a middle ground of `n_iter_sinkhorn = 500` in the rest of our experiments. We find that $t = 0.05$ gives the best trade-off between accuracy and traintime, especially at high dimension, and we thus adopt it as the default everywhere else.

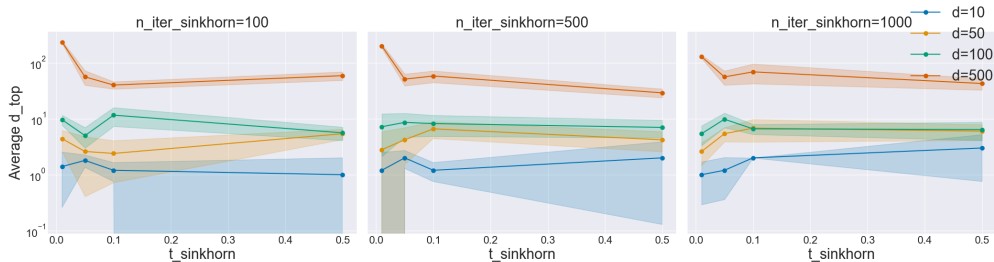

Figure 8: The figure illustrates the impact of key hyperparameters of the Sinkhorn operator on the performance of the DiffIntersort algorithm, as measured by the average toporder divergence ($D_{top}$). The relationship between the Sinkhorn temperature (`t_sinkhorn`, x-axis) and d_top (y-axis) is presented across three distinct settings for the number of Sinkhorn iterations (`n_iter_sinkhorn`), with each setting displayed in a separate subplot. Within each subplot, the effect of the number of variables (d) is shown, with each line representing a different dimensionality. The solid line denotes the mean $D_{top}$ averaged over all simulations, while the shaded region represents the standard deviation.

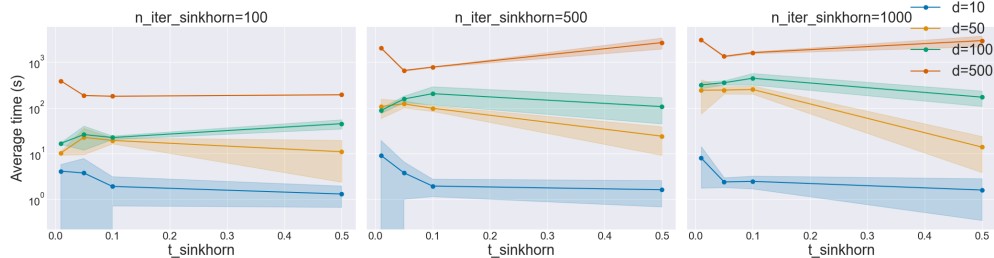

Figure 9: The figure displays the computational cost of the DiffIntersort algorithm, measured as the average execution time in seconds (y-axis), as a function of the Sinkhorn temperature (`t_sinkhorn`, x-axis). The analysis is partitioned into three subplots, each corresponding to a different number of Sinkhorn iterations (`n_iter_sinkhorn`). The scalability of the algorithm with respect to the number of variables (d) is demonstrated by the separate lines within each subplot. The solid line indicates the mean computation time, and the shaded area represents the standard deviation.

## J.5 SIMULATED DATA

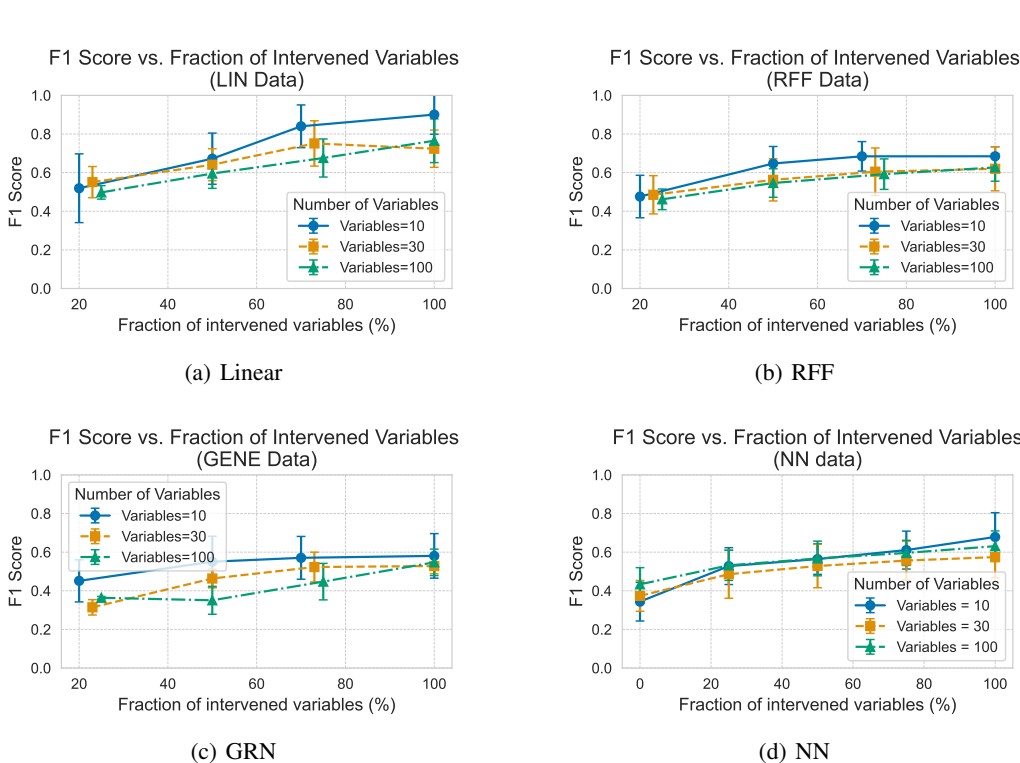

(a) Linear

(b) RFF

(c) GRN

(d) NN

Figure 10: F1 score of our algorithm with DiffIntersort constraint for the four considered data types over the fraction of intervened variables for 10, 30, and 100 variables. As can be observed, the performance is consistent across the scale of the number of variables as there is no major drop in performance at 100 variables compared to 10 and 30 variables.

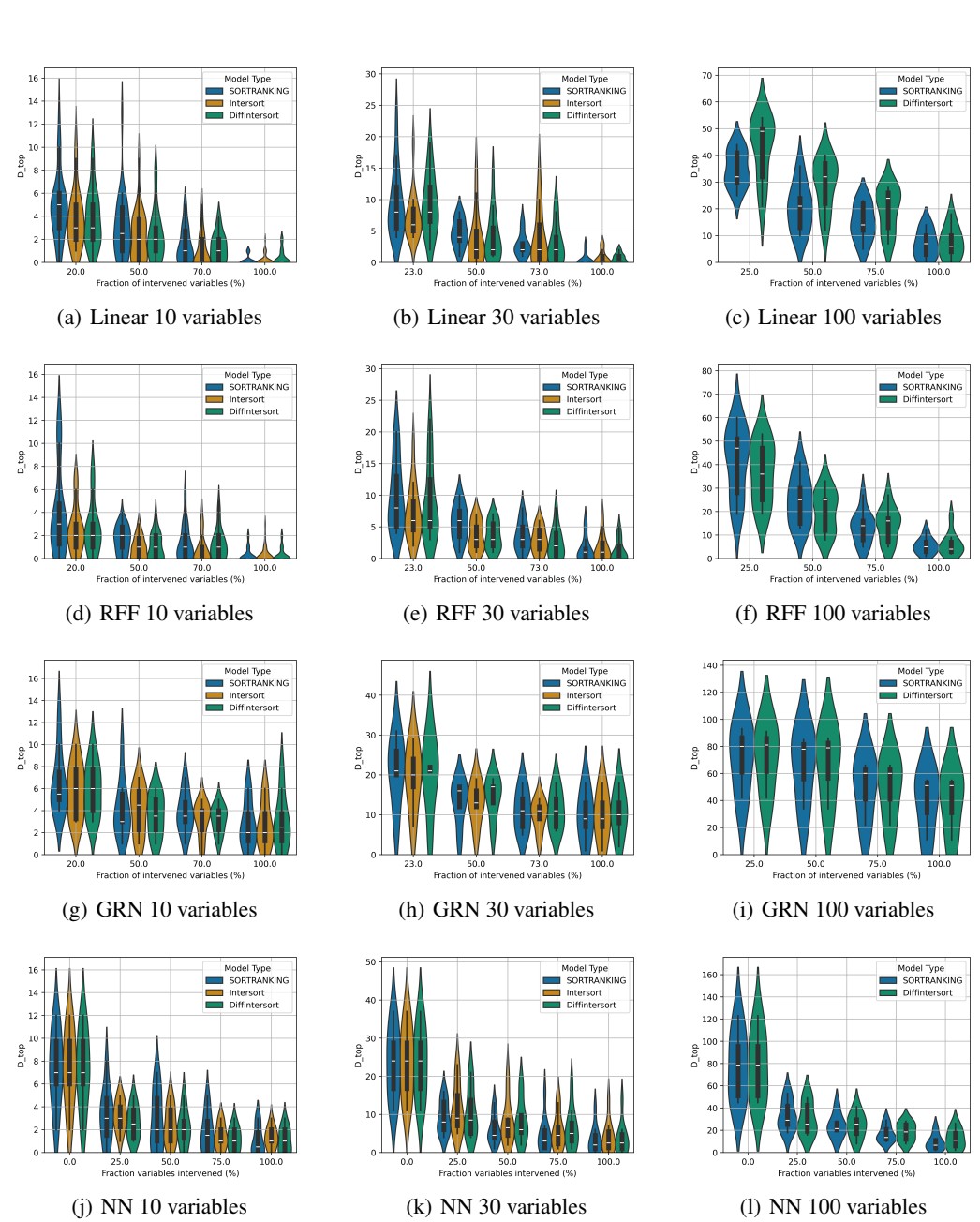

Figure 11: Top order diverge scores (lower is better) assessing the quality of the derived causal order, comparing our method based on the DiffIntersort score to SORTRANKING and Intersort on 10, 30 and 100 variables, for various types of data.

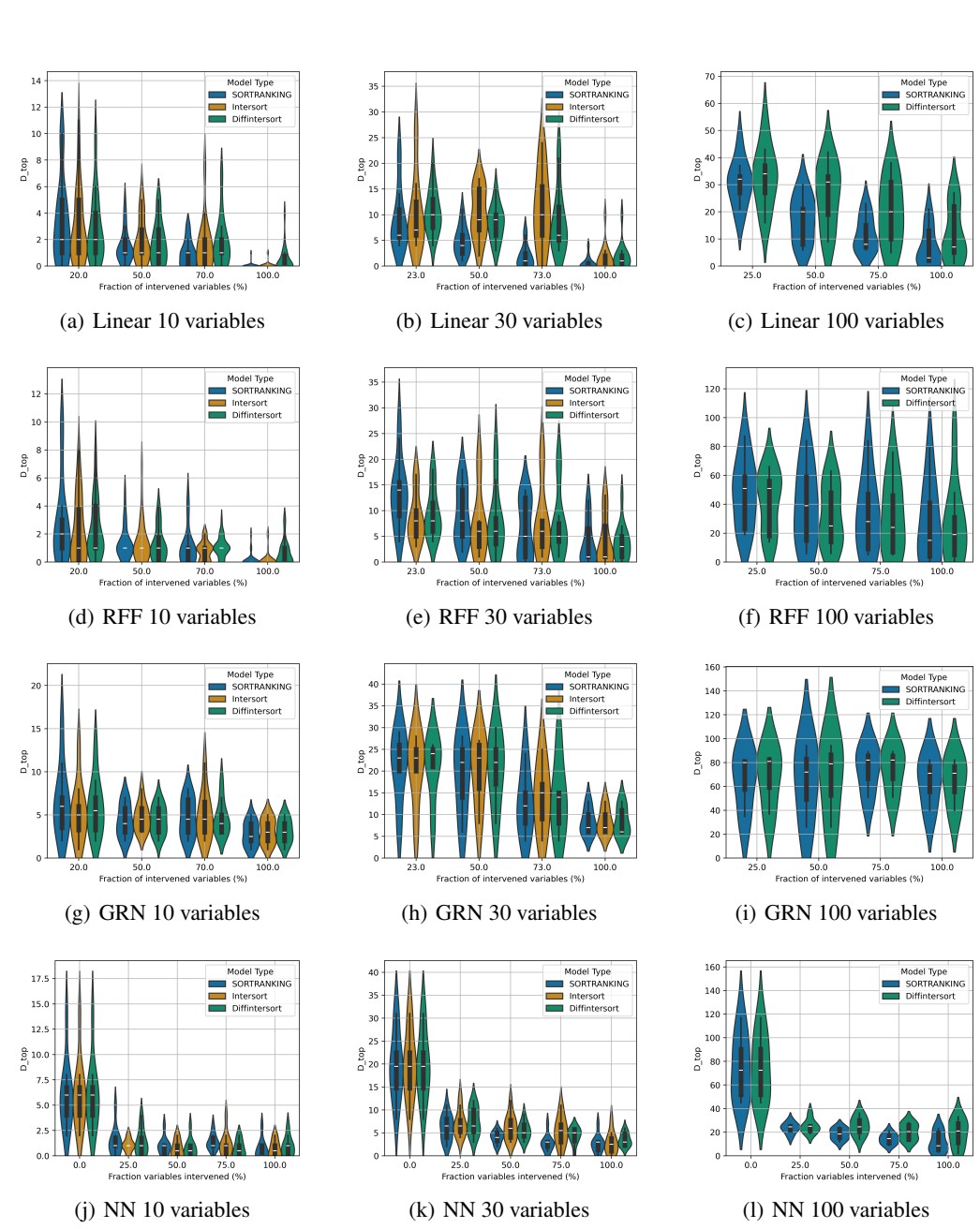

(a) Linear 10 variables   (b) Linear 30 variables   (c) Linear 100 variables

(d) RFF 10 variables   (e) RFF 30 variables   (f) RFF 100 variables

(g) GRN 10 variables   (h) GRN 30 variables   (i) GRN 100 variables

(j) NN 10 variables   (k) NN 30 variables   (l) NN 100 variables

Figure 12: Top order diverge scores (lower is better) assessing the quality of the derived causal order, comparing our method based on the DiffIntersort score to SORTRANKING and Intersort on 10, 30 and 100 variables, for various types of data for a scale free network distribution.

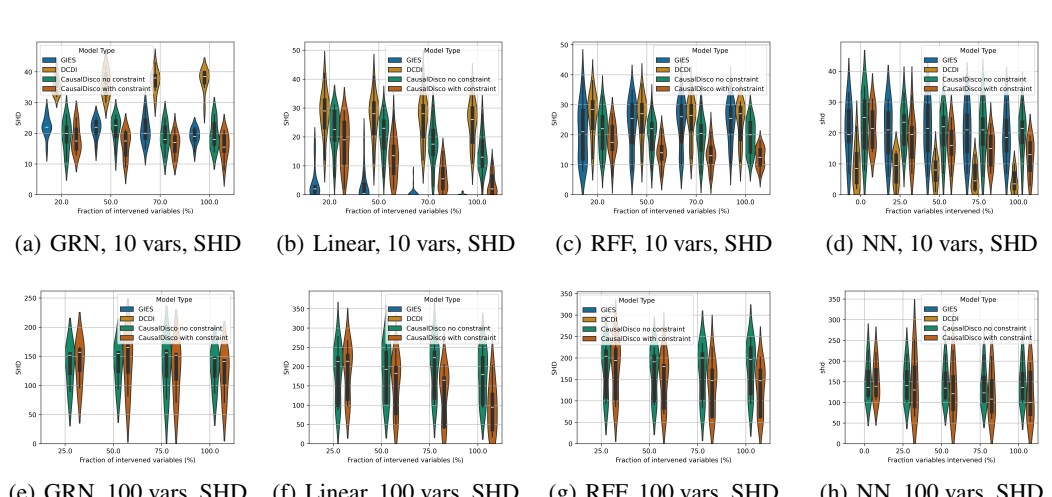

(a) GRN, 10 vars, SHD    (b) Linear, 10 vars, SHD    (c) RFF, 10 vars, SHD    (d) NN, 10 vars, SHD

(e) GRN, 100 vars, SHD    (f) Linear, 100 vars, SHD    (g) RFF, 100 vars, SHD    (h) NN, 100 vars, SHD

Figure 13: Comparison of Structural Hamming Distance (SHD) for Gene, Linear, RFF, and Neural Network models with varying numbers of variables.

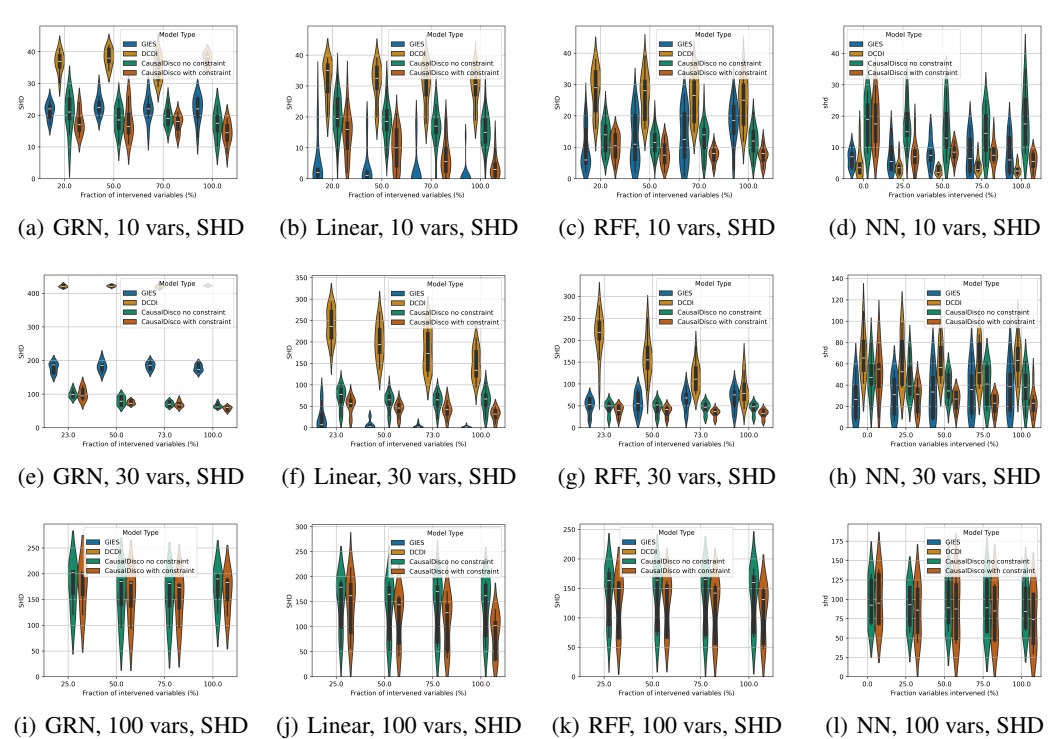

(a) GRN, 10 vars, SHD    (b) Linear, 10 vars, SHD    (c) RFF, 10 vars, SHD    (d) NN, 10 vars, SHD

(e) GRN, 30 vars, SHD    (f) Linear, 30 vars, SHD    (g) RFF, 30 vars, SHD    (h) NN, 30 vars, SHD

(i) GRN, 100 vars, SHD    (j) Linear, 100 vars, SHD    (k) RFF, 100 vars, SHD    (l) NN, 100 vars, SHD

Figure 14: Comparison of Structural Hamming Distance (SHD) for Gene, Linear, RFF, and Neural Network models with varying numbers of variables for a scale-free network distribution.

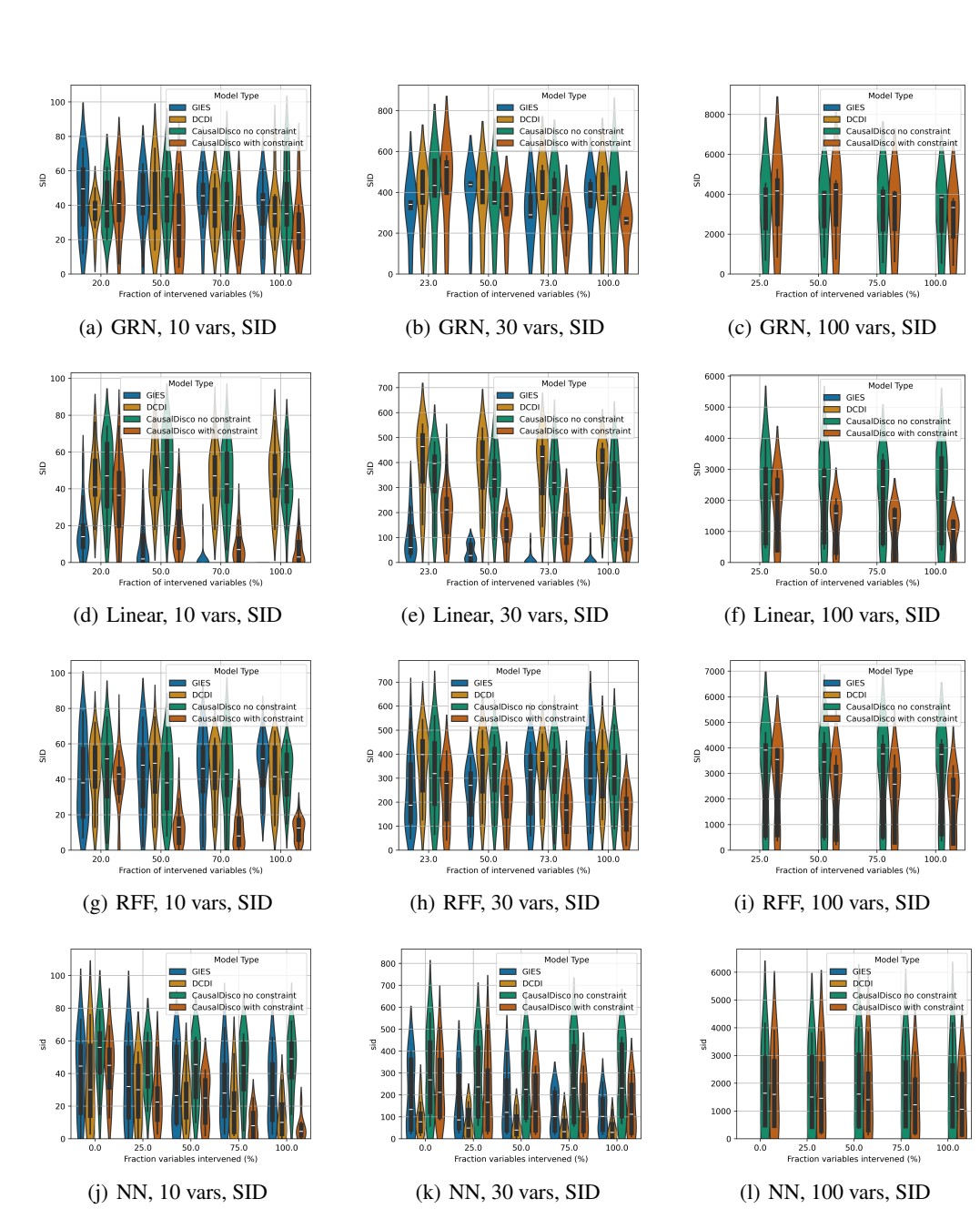

(a) GRN, 10 vars, SID     (b) GRN, 30 vars, SID     (c) GRN, 100 vars, SID

(d) Linear, 10 vars, SID     (e) Linear, 30 vars, SID     (f) Linear, 100 vars, SID

(g) RFF, 10 vars, SID     (h) RFF, 30 vars, SID     (i) RFF, 100 vars, SID

(j) NN, 10 vars, SID     (k) NN, 30 vars, SID     (l) NN, 100 vars, SID

Figure 15: Comparison SID (lower is better) for GRN, Linear, RFF, and Neural Network models with varying numbers of variables.

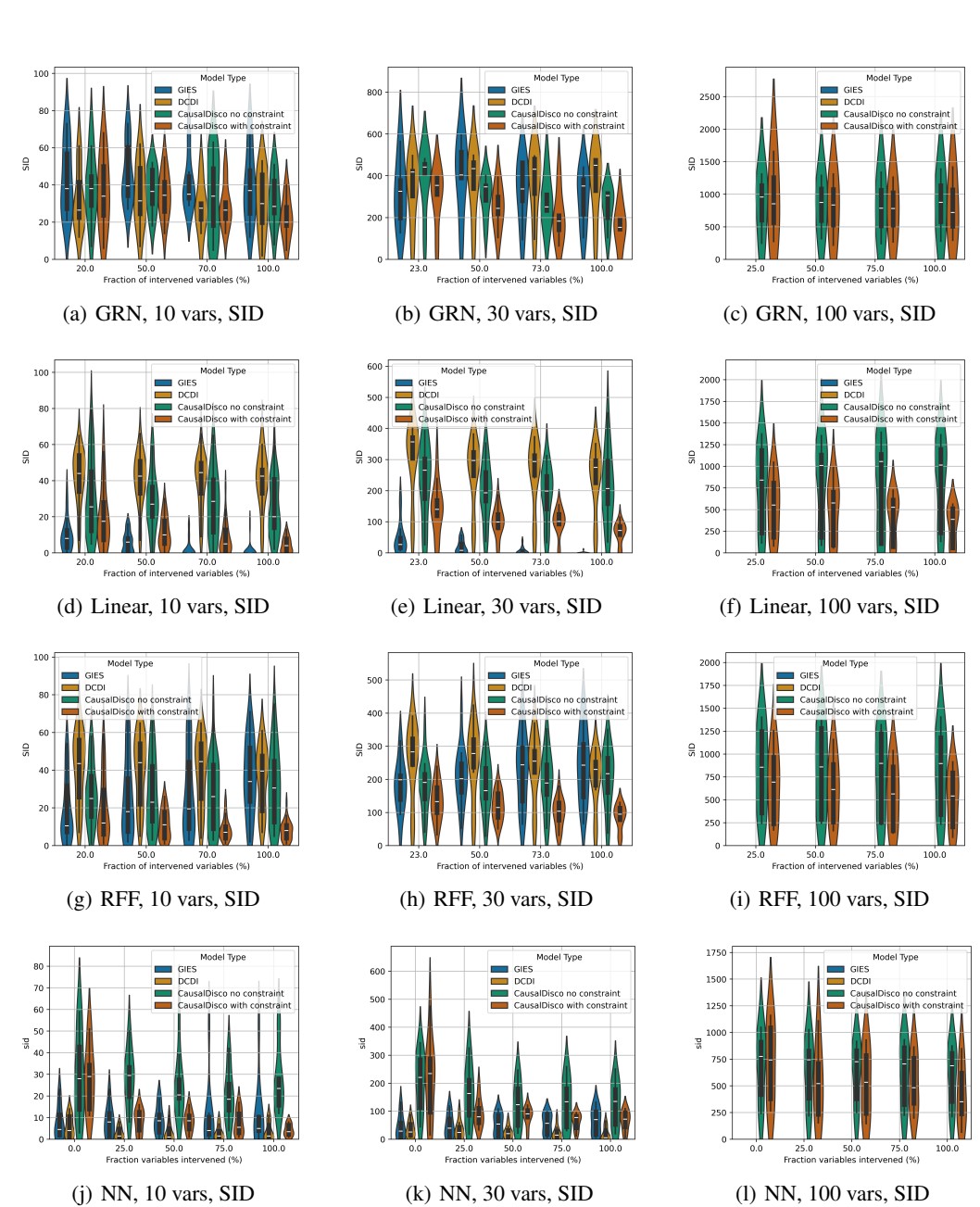

Figure 16: Comparison SID (lower is better) for GRN, Linear, RFF, and Neural Network models with varying numbers of variables for a scale-free network distribution.

### J.6 REAL-WORLD DATA EXPERIMENT: SACHS DATASET

To complement our synthetic evaluations, we conducted a small experiment on the protein-signaling dataset of Sachs et al. (2005), a widely used benchmark in causal discovery. This dataset consists of $n = 5846$ measurements of 11 phosphoproteins and phospholipids in human cells. Of these, 1755 are observational measurements, while the remaining 4091 are collected under five different perturbations, where reagents were applied to activate or inhibit specific proteins. Following common practice, we treat these perturbations as interventions and evaluate how well different methods recover the established consensus network of 17 edges.

We compare our causal discovery model with and without the DIFFINTERSORT regularizer. Structural Hamming Distance (SHD) to the consensus network is reported in Table 4, alongside published baselines results from Lorch et al. (2022).

Table 4: Performance on the Sachs et al. (2005) dataset. Lower SHD is better.

| Method | SHD |
| --- | --- |
| Our model w/ DiffIntersort | 14 |
| Our model w/o DiffIntersort | 17 |
| DCDI | 15 |
| GIES | 40 |

These results support our claim that the DiffIntersort regularizer improves recovery of causal structure in practice and achieves performance competitive with state-of-the-art approaches.

**Caveat.** As emphasized by Mooij & Heskes (2013), most perturbations in the Sachs dataset alter protein *activity* rather than *abundance*, meaning that they do not correspond precisely to atomic interventions in the sense of Pearl (2009). Therefore, care is needed in interpreting these results. We include this experiment as a proof of concept demonstrating compatibility of our approach with real-world pipelines, not as a definitive benchmark. Future work should assess the DiffIntersort regularizer on intervention datasets with well-validated mechanistic ground truth and on settings involving cycles or latent confounding.

## K COMPUTATIONAL RESOURCES

The experiments were run on a cluster with up to 20 CPUs, 8Gb of memory per CPU, and one GPU for DiffIntersort on large scale settings.

## L LLM USAGE DECLARATION

We declare that LLM systems were used to perform part of this work, such as polishing the text and presentation, writing (plotting) scripts, and searching related literature.

