# OpenReview forum: "Efficient Differentiable Discovery of Causal Order"
_ICLR.cc/2026/Conference — Submitted to ICLR 2026_

### Official Review · Reviewer_GVeX · 2025-10-26

**Soundness:** 2
**Presentation:** 3
**Contribution:** 2
**Rating:** 4
**Confidence:** 3

**Summary:**

This paper proposes a differentiable and scalable causal order regularizer based on Intersort (Chevalley et al. 2025c). Intersort is a score-based algorithm that discovers causal order using intervention data, but it relies on discrete combinatorial optimization, is computationally expensive, and is non-differentiable. By continuously relaxing the Intersort score using differentiable sorting techniques (specifically the Sinkhorn operator), this paper derives a differentiable surrogate objective (DiffIntersort). The authors repurpose this differentiable score as a regularizer to inject an inductive bias toward causal order into gradient-based learning systems. As a proof-of-concept, the authors apply this regularizer to a simple differentiable causal discovery model (called CausalDisco in the paper) and demonstrate on a variety of synthetic datasets (linear, RFF, GRN, and NN) that the inclusion of this regularizer significantly improves the model's structural recovery capabilities (SHD, SID) compared to the model without it, and surpasses baselines such as GIES and DCDI on certain nonlinear datasets.

**Strengths:**

1. The tech of a differentiable operator is novel to me and may provide insights into other fields.

2. The experiments are comprehensive.

**Weaknesses:**

1. This approach is very similar to DP-DAG (Charpentier et al. 2022) and BayesDAG (Annadani et al. 2023). DP-DAG is also permutation-based, differentiable (Sinkhorn), and supports intervening data. It is the approach's most direct and significant competitor. The lack of experimental comparisons with DP-DAG significantly reduces the persuasiveness of our empirical results, particularly in terms of scalability and accuracy.

2. Although the authors conduct experiments on RFF, GRN, and NN (non-linear) datasets, the paper does not explain how the model $f(\cdot)$ and the fitting loss $\mathcal{L}_{fit}$ are parameterized in these cases. This is a key missing implementation detail.

**Questions:**

above

---

> ### Author Response · Authors · 2025-11-20
>
> We thank the reviewer for the thoughtful comments and for highlighting both the comprehensive nature of the experiments and the potential broader impact of the differentiable operator. We are happy to clarify the two main points raised.
>
> 1. Relation to DP-DAG (Charpentier et al. 2022), BayesDAG (Annadani et al. 2023), and experimental baselines
>
> We appreciate the reviewer pointing out the conceptual connections. Indeed, these models and ours share the use of permutation-based relaxations via Sinkhorn.
>
> At the same time, the scope and goals of DiffIntersort differ substantially from these methods:
> - DP-DAG and BayesDAG are full causal discovery algorithms, optimizing likelihood- or ELBO-based objectives to learn DAG structures.
> - DiffIntersort is not proposed as a new causal discovery algorithm, but as a differentiable relaxation of the Intersort score tied to interventional faithfulness to infer the causal order, with new theoretical guarantees and a new use case as a causal-order regularizer.
>
> Because we are not proposing a new structural learner or claiming state-of-the-art performance, the experimental comparison focuses on:
> - Intersort vs DiffIntersort (discrete vs differentiable score), and
> - CausalDisco with vs without the DiffIntersort regularizer (to isolate the effect of the regularizer itself).
>
> In this context, the relevant question for a method like DP-DAG is not whether it performs better overall, but rather whether adding the DiffIntersort regularizer would help or hurt it. To avoid unnecessary complexity and confounding factors, we demonstrate this effect using a simple model (CausalDisco). This keeps the evaluation targeted to the contribution of DiffIntersort rather than to differences between entire causal discovery pipelines.
>
> 2. Parameterization of the model and the loss for nonlinear datasets
>
> A key aspect of the experiments is that the structural model $\mathcal{L}(\theta)$ is always the same linear SEM, regardless of whether the data were generated by linear, RFF, GRN, or NN mechanisms. The loss $\mathcal{L}_{fit}$ is therefore identical across all experiments and corresponds to the linear SEM reconstruction loss described in Section 2.3 and Appendix B.
>
> The nonlinearity in the experiments enters only through the data-generating mechanisms, not through the model parametrization. This is intentional: the purpose is to test whether the ordering regularizer still improves recovery even when the structural model is misspecified. This design isolates the effect of the DiffIntersort regularizer and avoids entangling it with more complex parameterizations.
>
> We will emphasize in the paper that the linear SEM is used in all settings and that the nonlinear datasets differ only in how the synthetic data are generated.
>
>
> We thank the reviewer again for the helpful feedback. We clarify the model/ loss parameterization in the nonlinear settings in the revised manuscript.

---

### Official Review · Reviewer_Fc3N · 2025-10-30

**Soundness:** 2
**Presentation:** 3
**Contribution:** 2
**Rating:** 4
**Confidence:** 3

**Summary:**

The paper aims to recover the correct causal order of a causal model with observational and single-variable interventional distributions. It extends the Intersort score (Chevalley et al. 2024), a metric for identifying the causal order, to a differentiable form using the Sinkhorn operator, thus enabling efficient computation. The method of using potential to represent causal order is similar to Annadani et al. (2023). The paper also proposes to use the Intersort score as a regularization to impose learning the causal structure.

**Strengths:**

- The development of scalable approaches is of interest in the field of causal discovery.

- Experiment shows the proposed method achieves efficient computation when the number of variables is large.

**Weaknesses:**

- The novelty is unclear to me, as the method resembles Annadani et al. (2023).
- The diffIntersort score is not convex, which might result in converging to a local minimum.
- The paper aims to improve computational efficiency with respect to the number of variables d. However, for enhancing the scalability of score-based causal discovery methods, it may be more important, in my view, to evaluate both accuracy and computational efficiency with respect to the dataset size instead. Especially, Intersort (Chevalley et al. 2024) has already achieved $O(d^3)$ computation complexity, which is not large.
- The section of diffIntersort as a regularizer is unclear, as illustrated in Questions.

**Questions:**

- When DiffIntersort score is utilized as a regularizer, does that make the loss function $\mathcal{L}_{fit}(\theta, p)$ non-differentiable?
- Conceptually, the correct model should maximize the DiffIntersort score. Why $\lambda > 0$ in equation (6)? Minimizing $\lambda S(p)$ would encourage worse causal model fitting.
- The linear SEM in equation (7) is confusing unless I miss some important points. If the observational distribution (and interventional distribution) is available, why can't $\boldsymbol{W}$ be inferred directly? The causal structure will be naturally given after we find that some $W_{j,i}$ is close to $0$.
- The analytical computational complexity of diffIntersort is $O(d^2 T)$, where $T$ is the iteration number that is typically $500$. It is unclear how this number is selected. Should there be a stopping criteria according to the convergence of $S(\boldsymbol{p})$?
- Is there any Bayesian interpretation for using the DiffIntersort score as a regularizer?

---

> ### Author Response · Authors · 2025-11-20
>
> We thank the reviewer for the constructive and thoughtful comments. We clarify the main points below and will further streamline the exposition in the revision.
>
> 1. Novelty and relation to Annadani et al. (2023)
>
> We appreciate the reviewer noting the conceptual similarity through the use of potentials and Sinkhorn relaxations. The novelty of our work is not in reintroducing these tools, but in applying them to a completely different causal framework:
> - Annadani et al. perform Bayesian posterior inference over DAGs.
> - DiffIntersort is a differentiable relaxation of the Intersort score, based on interventional faithfulness, which is neither likelihood-based nor Bayesian.
>
> Our contributions include:
>
> (1) a new continuous formulation of the Intersort score,
> (2) accompanying theoretical results (Lipschitz gradient, initialization guarantees), and
> (3) a new use of the score as a differentiable causal-order regularizer.
>
> 2. Non-convexity of the DiffIntersort score
>
> We agree that the relaxed score is non-convex. This is unfortunately very common in causal discovery. Ng et al. (2024), which we cite, discuss this phenomenon extensively.
>
> That said, DiffIntersort provides unusually strong optimization guarantees for this class of methods:
> - Theorem 2.6: gradient is Lipschitz continuous, ensuring stable gradient descent.
> - Proposition 2.8 and Theorem 2.9: the desirable solutions lie in a convex set, and
> - Lemma 2.10: SORTRANKING provides an initialization provably inside this convex set.
>
> These properties mitigate local minima issues compared to typical differentiable causal discovery models.
>
> 3. Scalability in $d$ rather than in sample size $n$
>
> All methods first compute a Wasserstein-based distance matrix; this is the only step influenced by $n$ and is shared across baselines. The subsequent optimization depends solely on $d$. Thus, scalability in the number of variables is the relevant axis. As shown in Figure 3, Intersort’s $O(d^3)$ scaling becomes prohibitive at large $d$, while DiffIntersort remains practical and GPU-friendly.
>
> 4. Using DiffIntersort as a regularizer
>
> Using the Sinkhorn relaxation ensures that the score remains differentiable, even though the overall loss is non-convex. This is standard for continuous relaxations in causal discovery and does not pose practical difficulties. If $\mathcal{L}_{fit}$ is differentiable, then the regularised loss remains differentiable.
>
> 5. Sign in Equation (6)
>
> We appreciate the reviewer pointing this out.
> Indeed, the correct formulation should maximize the DiffIntersort score, which in an additive loss means subtracting $S(p)$. This was a notational oversight, and we will correct it.
>
>
> 6. Why use a linear SEM? Why not infer W directly?
>
> Recovering W from observational and interventional data requires additional identifiability assumptions (e.g., noise independence, non-Gaussianity, equal residual variances), and even then, finite-sample estimation can be statistically challenging. Our goal in this section is not to propose a new SEM learner, but to provide a clean proof-of-concept showing that adding the DiffIntersort regularizer improves performance independently of the choice of structural model. The linear SEM isolates the contribution of the regularizer.
> Empirically, as shown on linear data, including the regularizer indeed improves recovery.
>
> 7. Bayesian interpretation
>
> Our method is not Bayesian, and we therefore avoid presenting a full Bayesian interpretation. However, using $S(p)$ as a regularizer is conceptually analogous to imposing a soft prior over causal orders, constraining the model toward structures that match interventional evidence.
>
> 8. Sinkhorn iterations and stopping criterion
>
> Appendix J.4 analyzes the choice of Sinkhorn iterations $T$ and its rationale. In our implementation we additionally apply early stopping once the matrix is sufficiently close to doubly stochastic.
>
>
> We thank the reviewer again for the insightful questions. We hope this clarifies your questions.

---

### Official Review · Reviewer_KpsU · 2025-11-03

**Soundness:** 2
**Presentation:** 1
**Contribution:** 3
**Rating:** 2
**Confidence:** 3

**Summary:**

The authors propose a method for causal order discovery based on interventional data that can be optimized through gradient descent. The work heavily builds on the INTERSORT method and provides a gradient-based optimization mechanism for the same score. A significant part of the contribution is the theory stating that the goal can be efficiently optimized. It is shown empirically that integrating DiffIntersort into the causal discovery method improves its performance.

**Strengths:**

1. The experimental evaluation is extensive.
2. The paper proposes a novel solution that significantly improves the scalability of previous methods.

**Weaknesses:**

1. The paper is hard to parse and is not self-contained:
* The notation is unclear (for example, in line 128, C in the upper index is not introduced).
* The assumptions, definitions, and theorems are informal and difficult to understand (for example, in assumption 2.3, what is a “detectable change”?; definitions 2.1 & 2.2 mix comments with the actual definitions; definition 2.7 is completely unclear to me)
* Some algorithms and theorems, which constitute a part of the proposed solutions, are not introduced properly: SORTRANKING, CausalDisco,  Theorems 2 & 4 from Chavalley at al. 2025c (line 348).
* Assumptions are not stated clearly. For example, in Assumption 2.3,  do the Authors assume single-node interventions?
* Section 2.2 could use additional structure to improve clarity.
2. The reproducibility of the results is low due to a lack of clear and detailed descriptions about the experimental setup and specifics about the proposed causal discovery method. This significantly reduces the potential impact and soundness of the work.

**Questions:**

1. Are the upper bounds from Thm2 and Thm4 upper-bounding the error or the accuracy of the method? Which method: Intersort, SORTRANKING, or both?
2. The CausalDisco method is not described. How does the proposed method for causal discovery work? Especially, how does it handle non-linear data and interventional data?
3. Figure 1 - What are the black lines (variance, standard error, or confidence intervals)? How were they computed? According to the caption, only one run was conducted in each configuration. Also, what are E and #edges? Which is the parameter of the setup, p_e or E, or #edges? Why not randomly select a certain number of variables for intervention for more reproducible results instead of sampling independently with probability p_int? Why does the performance of SORTRANKING often violate theoretical upper bounds?
4. Can DiffIntersort be used to regularize other causal discovery methods? Which ones?
5. What statements from section 2.2 apply to DiffIntersort? What are the theoretical guarantees for its convergence?
6. line 199 typo “finite t and T” -> “positive t and finite T”

---

> ### Author Response · Authors · 2025-11-20
>
> We thank the reviewer for the detailed feedback. We are happy to clarify the points below and will revise the paper for improved clarity and self-containment.
>
> 1. Clarity, Notation & Self-Containment
>
> We appreciate the comments regarding notation.
>
> - The SCM $\mathcal{C}$ is introduced in line 86 (“An SCM $\mathcal{C} = (\mathbf{S}, P_N)$…”) and all distributions $P_{X_j}^{\mathcal{C},.}$ denote the distributions induced by this SCM. We will restate this when first using the notation $P^{\mathcal{C}}$ to avoid any ambiguity.
> - “Detectable change” is formally defined in Definition 2.4, which states $D \left ( P_{X_j}^{\mathcal{C}, (\emptyset)}, P_{X_j}^{\mathcal{C}, do(X_i := {N}_i)} \right ) > \epsilon$. We will explicitly reference Definition 2.4 when introducing Assumption 2.3.
> - The formulation of Assumption 2.3 indeed assumes single-variable interventions, as stated in the text (“if intervening on variable X_i …”), and we are happy to make this even more explicit.
> - SORTRANKING is described in Appendix D, and CausalDisco is introduced in Section 2.3 with full details provided in Appendix B and Algorithm 1. We added a clarification that the linear SEM is also used for non-linear data.
> - For Theorems 2 & 4 from Chevalley et al. (2025c), we only use their numerical bounds as a sanity check (as noted in line 156). We are happy to recall their statements in the appendix for completeness (Appendix D).
>
> 2. Reproducibility
>
> We respectfully disagree that the reproducibility is low.
>
> We provide:
> - complete proofs in Appendix G,
> - complete data-generation details in Appendix H,
> - complete hyperparameters in Appendix I,
> - full pseudocode for the causal discovery method in Algorithm 1,
> - and the full code as supplementary material.
>
> These details are already described in the Reproducibility statement.
>
> 3. Questions
>
> Q1. Do the upper bounds from Thm 2 & 4 bound error or accuracy? And for which method?
>
> The bounds upper-limit the optimal order of the Intersort score (i.e., the error D_top of the optimal solution $\pi^*$; see line 156). Any method—SORTRANKING, Intersort, or DiffIntersort—should satisfy these bounds if their solution is close to the optimum. DiffIntersort consistently matches these bounds, whereas SORTRANKING may violate them because it is a heuristic that is not expected to reach the optimum.
>
> Q2. How does CausalDisco work? How does it handle nonlinear and interventional data?
>
> CausalDisco is introduced in Section 2.3 and described in full in Appendix B and Algorithm 1. It is deliberately kept simple—a linear SEM—so that the effect of the DiffIntersort regularizer can be isolated. Nonlinearity is present in the data-generating mechanisms, not in the SEM. Interventions are incorporated by masking the prediction loss of intervened variables (Appendix B.2). We can add a short summary of this in the main text to improve readability.
>
> Q3. Figure 1 details.
>
> Thank you for pointing this out.
>
> - The caption contains a typo: we ran 2 graphs per setting, not 1.
> - The black lines are 95% confidence intervals.
> - E denotes the set of edges (defined in line 80); #edges is the standard notation for its cardinality. The parameter of the setup is indeed $p_e$, as stated in the caption.
> - Sampling interventions i.i.d. with probability $p_{int}$  is chosen to match the theory (Chevalley et al., 2025c). For large d, this is equivalent to sampling a fixed number.
> - SORTRANKING can violate theoretical upper bounds because it is not guaranteed to reach the optimum of the score.
>
> These clarifications are included in the revision.
>
> Q4. Can DiffIntersort regularize other causal discovery methods?
>
> Yes, any method whose parameterization depends on the causal order. Appendix C.3 gives several concrete examples. We will emphasize this in the main text.
>
> Q5. Which theoretical statements apply to DiffIntersort? What guarantees exist?
>
> All statements in Section 2.2 that concern the continuous relaxation apply directly to DiffIntersort. In particular:
> - Theorem 2.6 establishes Lipschitz continuity of the gradient.
> - Proposition 2.8 shows convexity of the feasible initialization region.
> - Theorem 2.9 shows that all limiting optima lie in this region.
> - Lemma 2.10 shows that SORTRANKING provides an initialization in that region.
>
> Together, these results guarantee stable optimization and initialization near the global optimum of the discrete score.
>
> Q6. Typo at line 199.
>
> Thank you, we clarify that we meant “positive t” (temperature not tending to 0) and “finite T” (finite Sinkhorn iterations). We will update the wording for precision.
>
> We appreciate the reviewer’s recognition of the contribution (“novel solution”, “significantly improves scalability”, “extensive experiments”). The concerns raised relate primarily to presentation. We have addressed all points above and improved the clarity in the revised version.
> We hope these clarifications help convey that the method is principled, theoretically grounded, and fully reproducible.

---

### Meta-Review · Area_Chair_HdVh · 2026-01-05

**Summary:**

The reviewers have raised concerns primarily about clarity and relevance to the existing literature, especially regarding novelty. The criticisms pertain to the notation and the paper not being fully self-contained, citing unclear notation, informal definitions, and assumptions not explicitly stated. Furthermore, two reviewers have raised concerns about the novelty, particularly given the similarity of the approach to related permutation-based differentiable DAG methods (e.g., DP-DAG, BayesDAG, Annadani et al.).


While the authors have tried to provide clarifications in the revised manuscript and discuss their relevance to the existing literature, the scope of these clarifications and discussions is insufficient, rendering the paper still not ready for publication. I'd like to encourage the reviewers to provide detailed discussions of the similarities to existing perturbation-based approaches and to go beyond simply stating that the objectives are different. That may be true, but it still does not address the core question of the methodology's novelty. For instance, it is not clear whether the authors are repurposing a method designed by others for a different objective, or whether the mechanics are sufficiently distinct with novel components.

Overall, I do appreciate the authors' effort to bring more clarity to the manuscript. However, the overall amount of revisions needed to clarify the position and contribution of the paper is beyond what is done already and can be done between an original versus a final submission in the same conference.

**Reviewer Concerns:**

The authors have put effort into addressing the reviewer's concerns. While I appreciate the effort and I believe it addresses some of the concerns, I believe the manuscript still requires more discussion, especially on the connection to the existing literature.

**Reviewer Scores:**

I do not think the reviewers would have changed their ratings in any significant way.

---

### Decision · Program_Chairs · 2026-01-26

Reject